# HDAC1 dysregulation induces aberrant cell cycle and DNA damage in progress of TDP-43 proteinopathies

Cheng-Chun Wu[1,2,†], Lee-Way Jin[3,†], I-Fang Wang[1,†], Wei-Yen Wei[1], Pei-Chuan Ho[1], Yu-Chih Liu[1,2] & Kuen-Jer Tsai[1,2,4,*] [ID]

## Abstract

TAR DNA-binding protein 43 (TDP-43) has been implicated in frontotemporal lobar degeneration with ubiquitin-positive inclusions (FTLD-TDP) and amyotrophic lateral sclerosis. Histone deacetylase 1 (HDAC1) is involved in DNA repair and neuroprotection in numerous neurodegenerative diseases. However, the pathological mechanisms of FTLD-TDP underlying TDP-43 proteinopathies are unclear, and the role of HDAC1 is also poorly understood. Here, we found that aberrant cell cycle activity and DNA damage are important pathogenic factors in FTLD-TDP transgenic (Tg) mice, and we further identified these pathological features in the frontal cortices of patients with FTLD-TDP. TDP-43 proteinopathies contributed to pathogenesis by inducing cytosolic mislocalization of HDAC1 and reducing its activity. Pharmacological recovery of HDAC1 activity in FTLD-TDP Tg mice ameliorated their cognitive and motor impairments, normalized their aberrant cell cycle activity, and attenuated their DNA damage and neuronal loss. Thus, HDAC1 deregulation is involved in the pathogenesis of TDP-43 proteinopathies, and HDAC1 is a potential target for therapeutic interventions in FTLD-TDP.

**Keywords** DNA damage; FTLD; HDAC1; TDP-43
**Subject Categories** Cell Cycle; Neuroscience

## Introduction

Frontotemporal lobar degeneration (FTLD) and amyotrophic lateral sclerosis (ALS) are two of the most common neurodegenerative diseases in early-onset populations (age < 65 years). FTLD leads to impaired cognitive, language, and motor function (Irwin et al, 2015), whereas ALS is characterized by severe motor dysfunction with paralysis and respiratory failure to various extents (Chio et al, 2013). TAR DNA-binding protein 43 (TDP-43), a DNA/RNA-binding protein predominantly located in the nucleus, is highly conserved and ubiquitously expressed in eukaryotes (Cohen et al, 2011). Although its physiological function remains unclear, it has been generally implicated in multiple cellular functions, such as transcriptional repression, splicing, and translation (Buratti & Baralle, 2008; Wang et al, 2008; Lagier-Tourenne et al, 2010; Polymenidou et al, 2011). TDP-43 also pathogenically contributes to numerous neurodegenerative diseases, termed "TDP-43 proteinopathies", that are characterized by ubiquitin-immunoreactive cytosolic/nuclear inclusions containing misfolded TDP-43 in pathological samples. Such proteinopathies are identified by the presence of full-length TDP-43, polyubiquitinated TDP-43, phosphorylated TDP-43, and 35- and 25-kDa carboxyl TDP-43 fragments in the cytosolic/nuclear inclusions (Arai et al, 2006; Chen-Plotkin et al, 2010). To date, despite the high prevalence of FTLD-TDP and ALS, their underlying pathological mechanisms remain controversial, and no cures are available.

Recently, the involvement of aberrant cell cycle activity was identified in various disorders with progressive neurodegeneration or neuronal death, including Alzheimer's disease, ALS, Parkinson's disease, Huntington's disease, and ischemic stroke (Sharma et al, 2017). Normally, neurons in the adult central nervous system are terminally differentiated and remain quiescent. Thus, abnormal re-initiation of cell cycle activity and expression of cell cycle markers such as Ki-67 or proliferating cell nuclear antigen (PCNA) in mature neurons (Yang et al, 2001) might lead to cell apoptosis through p14[ARF]-, p53-, Bax-, and Apaf-1-mediated caspase-9 and caspase-3 activities or cell death by B-myb, C-myb, and BIM activities induced by E2F1 upregulation (Ranganathan & Bowser, 2003). In contrast, inhibitors of cyclin-dependent kinases or other cell cycle components can exert neuroprotective effects ameliorating this deregulation (Sanphui et al, 2013; Mi et al, 2016).

1  Institute of Clinical Medicine, College of Medicine, National Cheng Kung University, Tainan, Taiwan
2  Institute of Basic Medical Science, College of Medicine, National Cheng Kung University, Tainan, Taiwan
3  Department of Pathology and Laboratory Medicine, UC Davis Medical Center, Sacramento, CA, USA
4  Research Center of Clinical Medicine, National Cheng Kung University Hospital, College of Medicine, National Cheng Kung University, Tainan, Taiwan
*Corresponding author. Tel: +886 6 2353535 4254; Fax: +886 6 2758781; E-mail: kjtsai@mail.ncku.edu.tw
†These authors contributed equally to this work

The regulation of histone acetylation influences chromatin modulation and the transcription of genes that may be indispensable to normal biological processes such as cell proliferation and individual development (Graff & Tsai, 2013). Histone acetylation further modulates important brain functions, including neuronal differentiation and memory formation. In these critical aspects, histone deacetylases (HDACs) are regarded as essential regulators relevant to neurodegeneration. One member of this family, HDAC1, represses the transcription of specific cell cycle-related genes encoding such proteins as p21/WAF, E2F1, and cyclins A and E through interactions with their promoter regions (Brehm *et al*, 1998; Lagger *et al*, 2002; Rayman *et al*, 2002). A previous study showed that HDAC1 deregulation leads to aberrant cell cycle activity and neuronal cell death (Kim *et al*, 2008). Furthermore, HDAC1 is involved in responses to DNA damage and promotes the repair of DNA double-strand breaks (Miller *et al*, 2010; Wang *et al*, 2013; Gong *et al*, 2017).

We previously generated an FTLD-TDP mouse model that specifically overexpresses full-length TDP-43 in the forebrain under the control of the $Ca^{2+}$/calmodulin-dependent kinase II promoter (Tsai *et al*, 2010). These mice progressively exhibit cognitive deficits starting from 2 months of age and motor dysfunction accompanied by the downregulation of several markers of neuronal plasticity, neuronal loss, and hippocampal atrophy by 6 months of age. Their TDP-43 proteinopathies progress with age and with the presence of cytosolic polyubiquitinated TDP-43 and 35- and 25-kDa TDP-43 fragments in urea-soluble fractions. Thus, this model successfully mimics the pathogenesis of FTLD-TDP and can be used to investigate the mechanisms underlying neuronal death relevant to TDP-43 proteinopathies (Wang *et al*, 2012; Fang *et al*, 2014).

Little is known about the exact mechanism by which TDP-43 proteinopathies cause pathogenesis and neuronal death. We therefore examined whether cell cycle aberrance and DNA damage are involved in the degenerative progress of FTLD-TDP and investigated the role of HDAC1 in TDP-43 proteinopathies. Together, our findings illuminate the role of HDAC1 in TDP-43 proteinopathies and further support the hypothesis that restoring HDAC1 activity may be a feasible approach to treating FTLD-TDP and ALS.

# Results

## Aberrant cell cycle activity and DNA damage are involved in TDP-43 proteinopathies in FTLD-TDP Tg mice

Considering that aberrant cell cycle activity may be an essential pathogenic factor in various neurodegenerative conditions, we first investigated whether it is involved in the pathogenesis of TDP-43 proteinopathies such as FTLD-TDP. Immunofluorescence (IF) staining of Ki67 and TDP-43 in the frontal cortices and hippocampus revealed that Ki67 immunoreactivity became greater during the onset of FTLD-TDP Tg mice at the age of 6 months than in age-matched wild-type (WT) mice. Simultaneously, TDP-43 immunoreactivity was mislocalized to the cytosol, which caused reduced nuclear levels (Fig 1A, upper graph) and revealed significant pathological changes in TDP-43 proteinopathies (Neumann *et al*, 2006). Moreover, triple staining of Ki67, TDP-43, and NeuN (Fig 1A, lower graph) indicated that most of the cells with TDP-43 mislocalization

were neurons (68.9 ± 5.5%). Similarly, most of the Ki67-positive signals were occurred in neurons of FTLD-TDP Tg mice (66.1 ± 4.4%). Quantified results from the frontal cortices indicated that the number of cells or neurons with aberrant cell cycle activity was significantly increased in FTLD-TDP Tg mice (Fig 1B) and strongly correlated with TDP-43 proteinopathies, since aberrant cell cycle activity cannot be detected without TDP-43 mislocalization at the age of 2 months of FTLD-TDP Tg mice (Fig EV1), which imply that TDP-43 proteinopathies precede the onset of cell cycle aberrance. Subsequently, to confirm this cell cycle dysregulation in our FTLD-TDP mouse model, we evaluated cell cycle-related genes and proteins, including E2F1, cyclins A and E, PCNA, and p21 in the frontal cortices and hippocampus from 6 months old of FTLD-TDP or WT mice using reverse transcription–polymerase chain reactions (Fig 1C, left panel) and Western blotting (Fig 1C, right panel). The resulting data showed that these genes and proteins were upregulated in FTLD-TDP Tg mice, which suggest that aberrant cell cycle activity is involved in TDP-43 proteinopathies. These findings indicate that cell cycle dysregulation in neurons is a pathogenic feature in FTLD-TDP-associated neurodegeneration.

Next, to evaluate whether TDP-43 proteinopathies induce DNA damage in FTLD-TDP, we performed a comet assay to detect DNA double-strand breaks in the frontal cortices and hippocampus. The resulting data showed that 6-mon-old FTLD-TDP Tg mice possessed more cells with DNA double-strand breaks than age-matched WT mice did (Fig 2A). Furthermore, we investigated the relationship between TDP-43 proteinopathies and DNA damage with IF staining of TDP-43 and γH2AX in 6-mon-old FTLD-TDP Tg and WT mice (Fig 2B, upper graph). The quantified results for the frontal cortices revealed a significant increase in γH2AX-positive cells/neurons in the frontal cortices of FTLD-TDP Tg mice (Fig 2B, lower panel), and this abnormality was strongly associated with TDP-43 proteinopathies and cannot be detected in other unaffected brain region (Appendix Fig S1). In addition, we further found that neurons with mislocalized TDP-43 accounted for 70 ± 6.3% of the γH2AX-positive cells in frontal cortices and hippocampus of FTLD-TDP Tg mice, which suggest that DNA damage strongly involved in the pathogenesis of TDP-43 proteinopathies, especially for neurons, in the brains of FTLD-TDP mice.

Furthermore, IF staining of Ki67 and γH2AX (Fig 2C, upper graph) in the frontal cortices and hippocampus of 6-mon-old FTLD-TDP Tg and WT mice further revealed that cells/neurons with aberrant cell cycle activity also suffered DNA damage, which suggests that both pathological features are involved in TDP-43 proteinopathies. By further evaluating the percentage of neurons among cells that were both Ki67-positive and γH2AX-positive in FTLD-TDP Tg mice, we confirmed that neurons were the major damaged cell type co-exhibiting cell cycle aberrance and DNA damage in the frontal cortices and hippocampus of 6-mon-old FTLD-TDP Tg mice (Fig 2C, lower panel). This reinforces the hypothesis that cell cycle aberrance and DNA damage are highly related to neuronal degeneration during the pathogenesis of TDP-43 proteinopathies.

## FTLD-TDP Tg mice show reduced nuclear HDAC1 levels and activity

The function of HDAC1 in modulating cell cycle activity and DNA repair in the brain is well characterized, as the dysregulation of

HDAC1 is seen in various neurodegenerative conditions (Kim et al, 2008; Miller et al, 2010). Having confirmed the involvement of cell cycle aberrance and DNA damage in neuronal degeneration in FTLD-TDP Tg mice, we further sought to investigate the role of HDAC1 in the pathogenesis of TDP-43 proteinopathies. We first found that the 6 months old of WT and FTLD-TDP Tg mice did not differ in the HDAC1 levels measured in fractional extracts of frontal cortices and hippocampus obtained with radioimmunoprecipitation assay (RIPA) buffer (Fig 3A), but the FTLD-TDP Tg mice exhibited significantly higher HDAC1 levels in their cytosol than in their cell nuclei (Fig 3B). We further observed that HDAC1 was exported from the nucleus to the cytosol in FTLD-TDP Tg mice and co-localized with SMI32-immunoreactivity, which implies that HDAC1 was aberrantly modulated and mislocalized within the subcellular

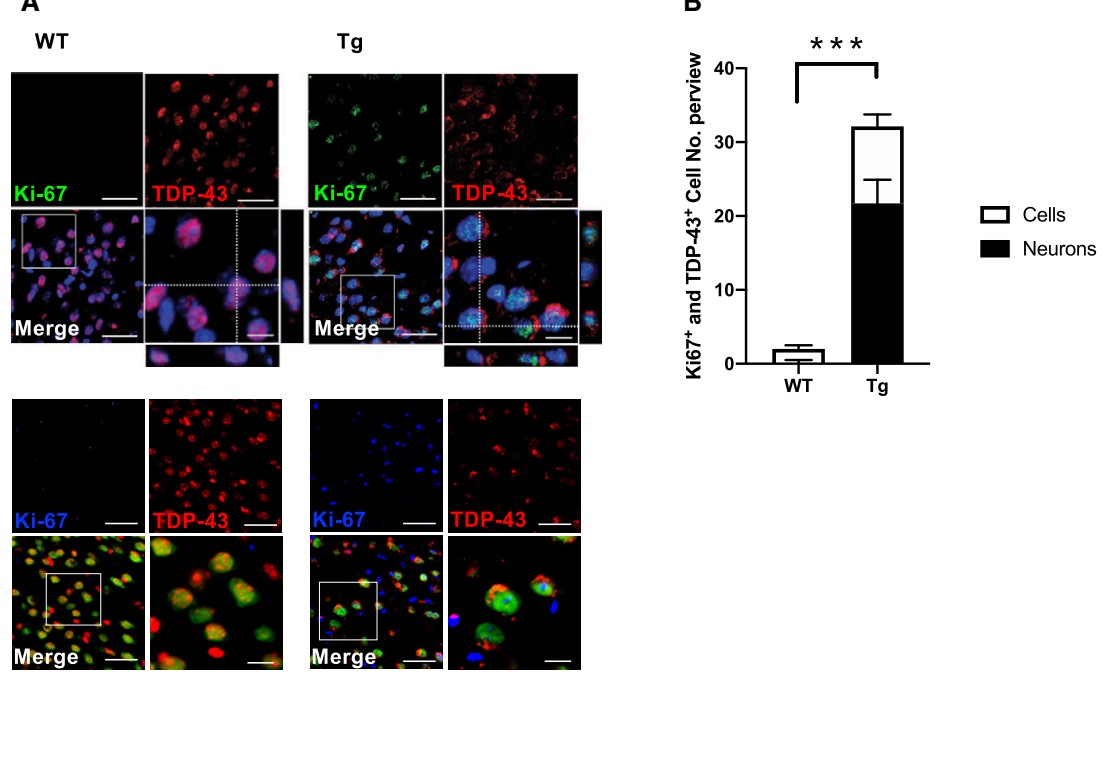

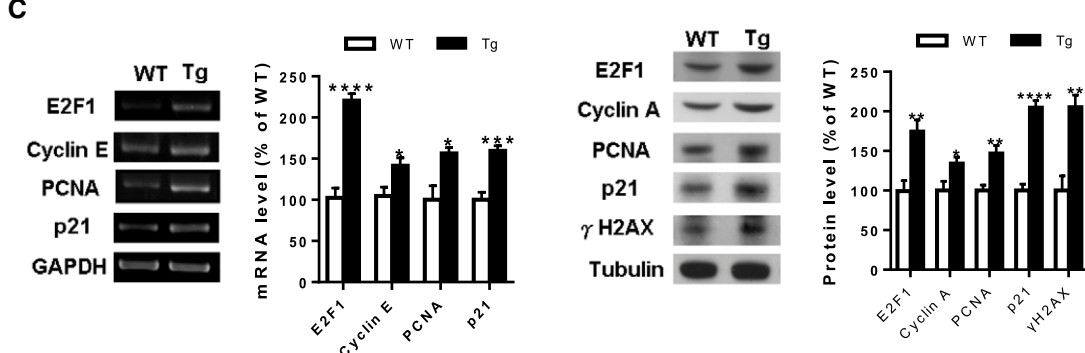

**Figure 1. Aberrant cell cycle activity correlates with TDP-43 proteinopathies in FTLD-TDP Tg mice.**

A   Representative immunofluorescence (IF) staining of Ki67 and TDP-43 in the regions of frontal cortices from 6-mon-old WT and FTLD-TDP Tg mice. Nuclei were stained with 4′,6-diamidino-2-phenylindole (DAPI; upper panel in blue) or neural marker NeuN (lower panel in green). Scale bar: 50 μm. The circled area is emphasized for showing the distribution of immunoreactivity in cell subregions. Scale bar: 15 μm.

B   Quantification of cells or neurons with Ki67 immunoreactivity and TDP-43 mislocalization from each view of microscope. $n = 9$ sections per mouse, $N = 5$ mice per group, data are presented as mean $\pm$ SEM, ***$P = 0.0007$ by t-test.

C   Representative data of reverse transcription PCR (left panel) or Western blot (right panel) for cell cycle-related genes and semi-quantification of the expression levels in the frontal cortices and hippocampus from the 6-mon-old WT and FTLD-TDP Tg mice. $N = 5$ mice per group, data are presented as mean $\pm$ SEM (%), statistical analysis by multiple t-test with FDR correction, Q = 1%. *$P < 0.05$, **$P \leq 0.01$; ***$P \leq 0.001$ and ****$P \leq 0.0001$, exact P values are shown in Appendix Table S1.

Source data are available online for this figure.

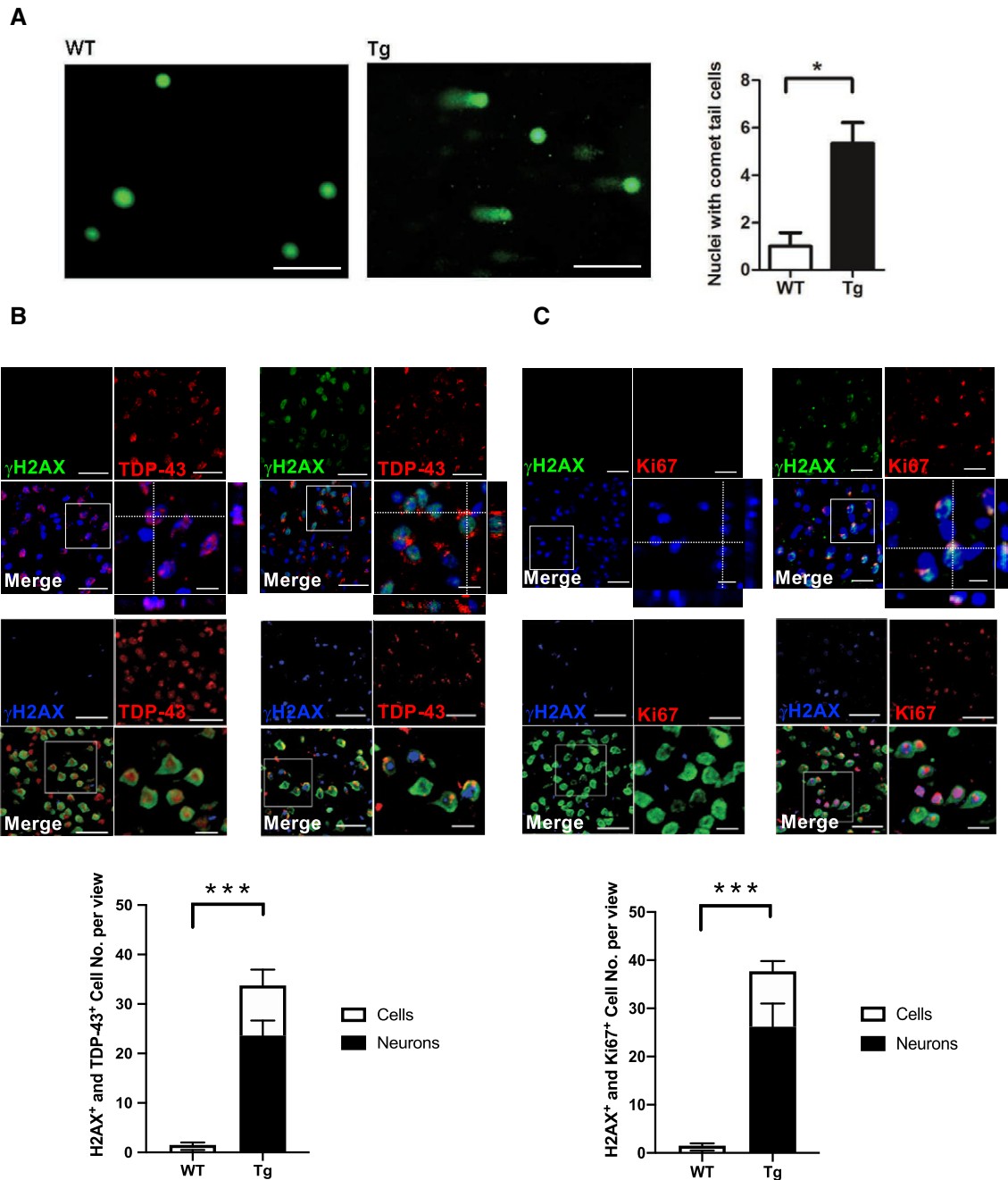

**Figure 2. DNA damage correlates with TDP-43 proteinopathies in FTLD-TDP Tg mice.**

A   Representative image of comet assay for DNA fragmentation and the quantification of cells with comet tails in the regions of frontal cortices from 6-mon-old WT and FTLD-TDP Tg mice. Scale bar: 50 μm. n = 9 sections per mouse, N = 5 mice per group, data are presented as mean ± SEM, *P = 0.0114 by t-test.

B   Representative IF staining of γH2AX and TDP-43 in the regions of frontal cortices from 6-mon-old WT and FTLD-TDP Tg mice. Nuclei were stained with DAPI (upper panel in blue) or NeuN (middle panel in green). Scale bar: 50 μm. The circled area is emphasized for showing the distribution of immunoreactivity in cell subregions. Scale bar: 15 μm. Lower panel: quantification of cells or neurons with γH2AX immunoreactivity and TDP-43 proteinopathies from each view of microscope. n = 9 sections per mouse, N = 5 mice per group, data are presented as mean ± SEM, ***P = 0.0008 by t-test.

C   Representative IF staining of γH2AX and Ki67 in the regions of frontal cortices from WT and FTLD-TDP Tg mice. Nuclei were stained with DAPI (upper panel) or NeuN (middle panel). Scale bar: 50 μm. Subregions, scale bar: 15 μm. Lower panel: quantification of cells or neurons with γH2AX and Ki67 immunoreactivity from each view of microscope. n = 9 sections per mouse, N = 5 mice per group, data are presented as mean ± SEM, ***P = 0.0004 by t-test.

Source data are available online for this figure.

compartments, thus revealing the nature of the neuronal damage (Werner et al, 2001; Kim & Casaccia, 2010) (Fig 3C). In addition, we investigated nuclear HDAC1 activity and the level of its potential targets, including P21 (Zupkovitz et al, 2010), E2F1 (Mitsiades et al, 2004), and potential substrate acetyl-histone H3 (Lys 9/14) (Dovey et al, 2010; Ferreira et al, 2017), to validate the catalytic function of HDAC1 in FTLD-TDP Tg mice at age of 6 months. Our results showed that nuclear HDAC1 activity was reduced in FTLD-TDP Tg mice (Fig 3D), whereas its targets and substrate levels were increased (Figs 1C and 3E), which suggest that nuclear HDAC1 function is aberrantly lost during pathogenesis in FTLD-TDP Tg mice.

## HDAC1 mislocalization correlates with pathogenesis of TDP-43 proteinopathies

To clarify the pathogenic role of HDAC1 deregulation in FTLD-TDP Tg mice, we examined the association between HDAC1 mislocalization and TDP-43 proteinopathies in the progression of FTLD-TDP. IF staining of HDAC1 and TDP-43 in the frontal cortices and hippocampus of FTLD-TDP Tg mice at the ages of 1, 6, and 12 months (Fig 4A, left graph) indicated that HDAC1 and TDP-43 were co-localized in the cell nuclei in the frontal cortices of 1-mon-old FTLD-TDP Tg mice. However, at the age of 6 months, when the

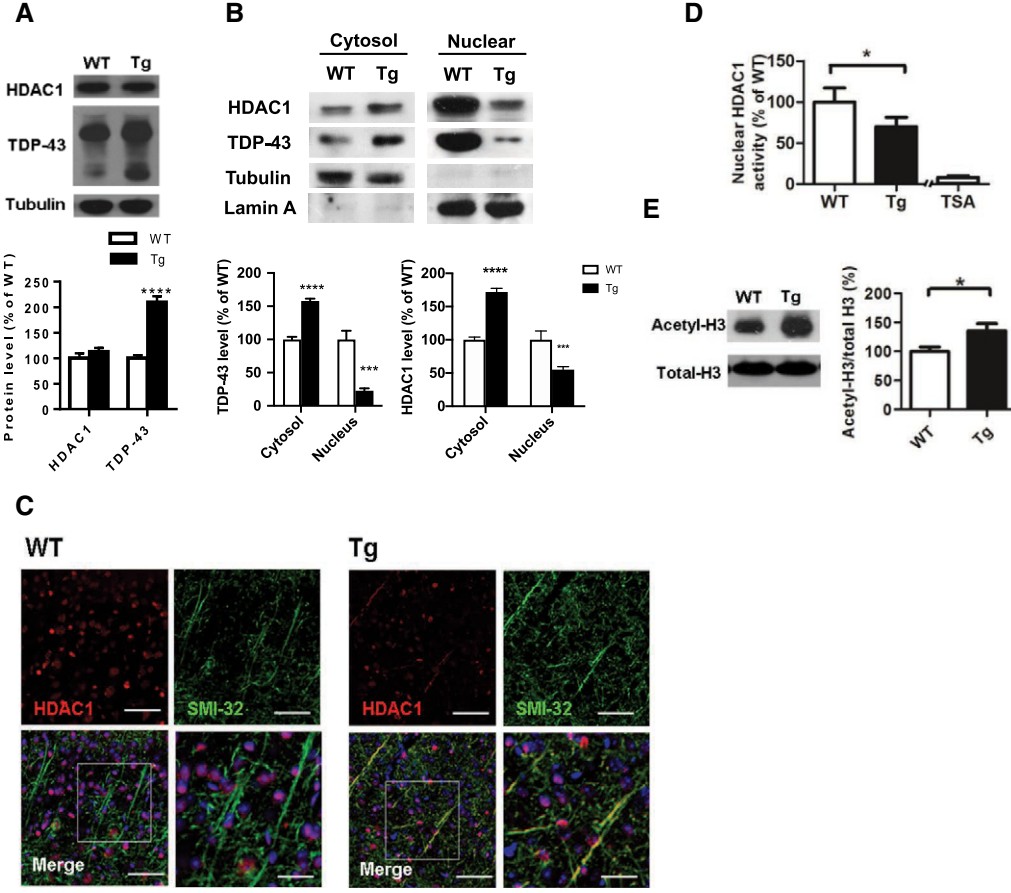

**Figure 3. HDAC1 dysregulation in FTLD-TDP Tg mice.**

A Top, representative Western blot data of HDAC1 and TDP-43 in extracts obtained following RIPA fractional extraction in the frontal cortices and hippocampus from 6 months old of FTLD-TDP or WT mice. Bottom, semi-quantification of HDAC1 and TDP-43 expression levels. N = 5 mice per group, data are presented as mean ± SEM (%), ****P < 0.0001 by multiple t-tests.

B Top, representative Western blot data of HDAC1 and TDP-43 in the cytosolic and nuclear extracts. Bottom, quantification of HDAC1 and TDP-43 levels. N = 5 mice per group, data are presented as mean ± SEM (%), ***P = 0.0005 (TDP-43) or 0.0004 (HDAC1); ****P < 0.0001 by multiple t-tests.

C IF staining of HDAC1 and SMI-32 in the regions of frontal cortices from WT and FTLD-TDP Tg mice. Scale bar: 150 μm. Blocked area in left bottom picture is showed in a magnified view in the right bottom picture. Scale bar: 50 μm.

D Nuclear HDAC1 activity assay from frontal cortices and hippocampus 6 months old of FTLD-TDP and WT mice. TSA: nuclear extracts that were treated with Trichostatin A (TSA, an HDAC inhibitor) as a negative control for HDAC1-transferred fluorescent activity during the HDAC1 activity assay. N = 5 mice per group, data are presented as mean ± SEM (%), *P = 0.0415 by t-test.

E Left, representative Western blot data of nuclear acetylated-histone H3 (Lys 9/14) and total histone H3 in 6 months old of WT and FTLD-TDP Tg mice. Right, quantification of nuclear acetylated-histone H3/total histone H3 ratio. N = 5 mice per group, data are presented as mean ± SEM (%), *P = 0.0147 by t-test.

Source data are available online for this figure.

impairments of the Tg mice become more pronounced (Tsai *et al*, 2010), we observed obvious co-mislocalization of TDP-43 and HDAC1 from the nucleus to the cytosol, which corresponded to the pathogenesis of TDP-43 proteinopathies. At the age of 12 months, the co-mislocalization of TDP-43 and HDAC1 had progressed in the

cells of FTLD-TDP Tg mice but not in the cells of age-matched WT mice, which revealed an age-dependent effect (Fig 4A, right histogram). We also confirm that more γH2AX expressed in the nucleus when cells undergo HDAC1 mislocalization in the frontal cortex of 12-month-old FTLD-TDP Tg mice (Fig 4B, left graph), but not in the

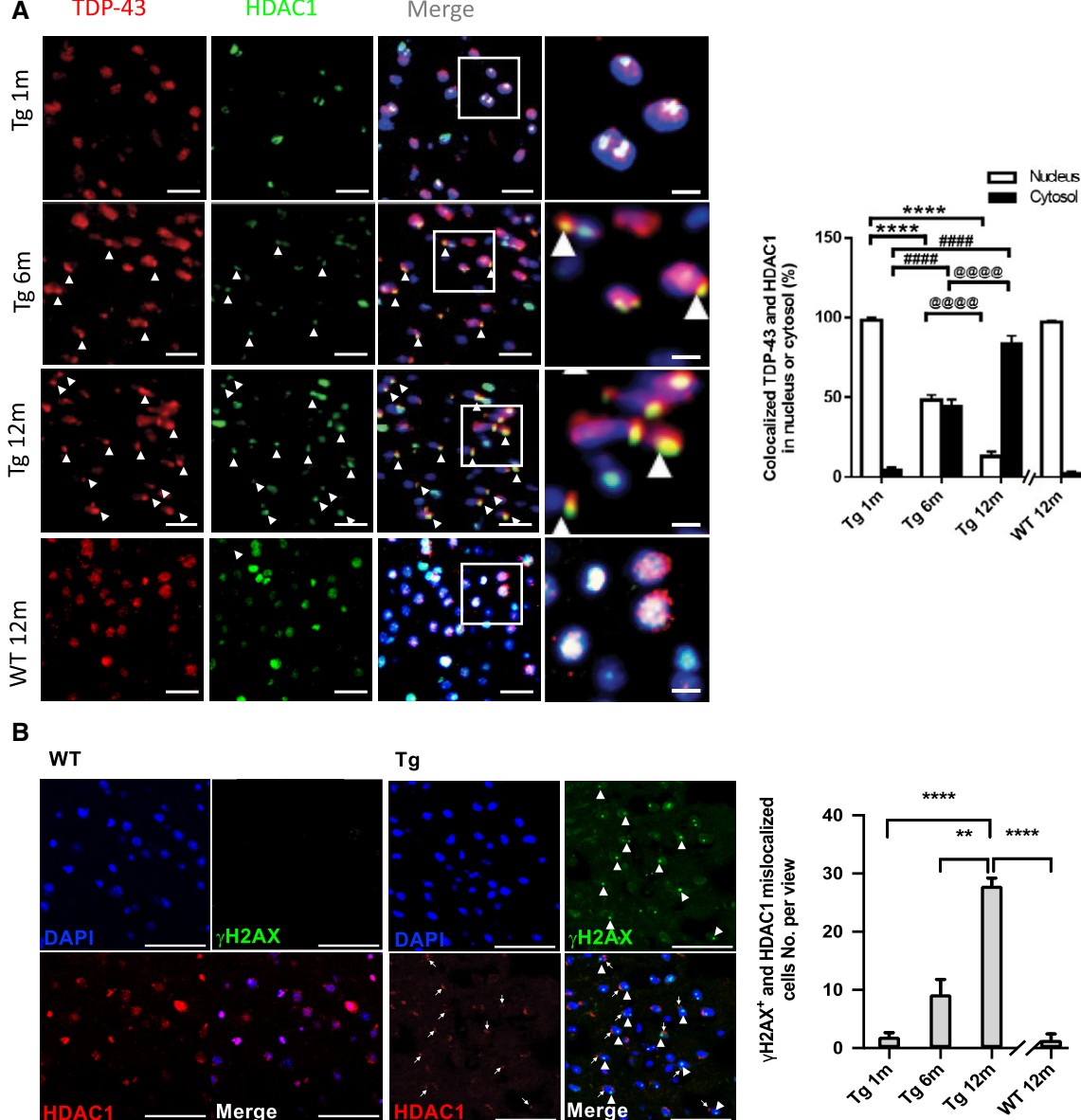

**Figure 4. HDAC1 mislocalization correlates with pathogenesis of TDP-43 proteinopathies in FTLD-TDP Tg mice.**

A   Left graph: IF staining of TDP-43 and HDAC1 during progression of TDP-43 proteinopathies in the frontal cortices from the FTLD-TDP Tg and WT mice. Nuclei were stained with DAPI (in blue). Scale bar: 50 μm. The circled area is emphasized for showing the distribution of immunoreactivity in cell subregions. Scale bar: 15 μm. Right histogram: Quantification of co-localized TDP-43 and HDAC1 immunoreactivity in the cytosol or nucleus in the WT or 1-, 6-, and 12-month-old FTLD-TDP Tg mice. n = 9 sections per mouse, N = 5 mice per group, data are presented as mean ± SEM. *Nucleus: Tg 1 m versus Tg 6 m or 12 m; #Cytosol: Tg 1 m versus Tg 6 m or 12 m; @ Tg 6 m versus Tg 12 m. ****/####/@@@@ *P* < 0.0001 by multiple comparison.

B   Left graph: IF staining of γH2AX and HDAC1 in the frontal cortices from the 12-month-old FTLD-TDP Tg and WT mice. Nuclei were stained with DAPI (in blue). Arrowhead: γH2AX+ nucleus; arrow: mislocalized HDAC1. Scale bar: 50 μm. Right histogram: Quantification of γH2AX and HDAC1 mislocalized cells in the WT or 1-, 6-, and 12-month-old FTLD-TDP Tg mice. n = 4 sections per mouse, N = 5 mice per group, data are presented as mean ± SEM. **P = 0.0079, ****P < 0.0001 by multiple comparison.

Source data are available online for this figure.

cells of age-matched WT mice (Fig 4B, right histogram). These data suggest that HDAC1 deregulation and DNA damage are strongly associated with the progression of TDP-43 proteinopathies.

## TDP-43 interacts with HDAC1 and traps it in cytosolic inclusions during the pathogenesis of TDP-43 proteinopathies

To investigate the mechanism by which TDP-43 proteinopathies affect HDAC1 mislocalization, we examined the HDAC1-TDP-43 interaction using an *in vitro* model, i.e., 293T cells overexpressing flag-tagged HDAC1 and myc-tagged TDP-43, followed by immunoprecipitation and immunoblotting. An HDAC1-TDP-43 interaction was detected in the co-transfected cell lysates (Fig 5A). With protein structural mapping, we further found that HDAC1 interacted with TDP-43 via the N-terminal region within the HDAC catalytic domain (Fig 5B). Furthermore, by immunoprecipitation of HDAC1 in cytosolic fractions of 6-mon-old FTLD-TDP Tg mice, we identified a consistent HDAC1-TDP-43 interaction even after export from the nucleus to the cytosol (Fig 5C), which implies an association between TDP-43 proteinopathies and HDAC1 mislocalization. We thus speculated that HDAC1 was trapped in TDP-43 inclusions during pathogenesis and further investigated this by testing for HDAC1 and TDP-43 in urea-soluble fractions. The HDAC1 levels in urea-soluble fractions from 6-mon-old FTLD-TDP Tg mice were significantly higher than those in urea-soluble fractions from age-matched WT mice (Fig 5D). Together, these results confirm the strong association between HDAC1 function loss and TDP-43 proteinopathies in the pathogenesis of FTLD-TDP. TDP-43 proteinopathies may play an essential role in the reduced nuclear levels and activity of HDAC1, and this deregulation may induce aberrant cell cycle activity and DNA damage during disease progression in FTLD-TDP.

## Increased HDAC1 activity ameliorates the cognitive and motor function impairments, aberrant cell cycle activity, DNA damage, and neuronal loss in FTLD-TDP Tg mice

Our findings raised the possibility that TDP-43 proteinopathies cause both cell cycle aberrance and DNA damage by reducing nuclear HDAC1 levels and activity. Consequently, to develop a therapeutic approach targeting nuclear HDAC1 deregulation, we first evaluated the therapeutic feasibility of manipulating the activity of HDAC1 in FTLD-TDP mice. To this end, a potential non-selective class 1 HDAC activator, theophylline (Ito *et al*, 2002; Cosio *et al*, 2004), was used to improve the function of class 1 HDACs.

Intriguingly, the behavioral outcomes data indicated that theophylline treatment ameliorated cognitive and motor function deficits in the Morris water maze and rotarod tests following 2 months of treatment (Fig EV2A and B). Theophylline also restored nuclear HDAC1 activity in the FTLD-TDP Tg mice (Fig EV2C), which implies that improving nuclear HDAC1 function, is potential for reversing TDP-43 proteinopathies in these mice.

Therefore, we sought to further investigate the therapeutic efficacy of specifically increasing HDAC1 activity by using a synthetic compound of HDAC1-specific activator called 5104434 (ChemBridge Corporation, San Diego, CA). Its specificity and effectiveness were described in a published international patent by Tsai *et al* (2010) (Patent No. WO2010011318). To further confirm its therapeutic potential before *in vivo* treatment, we examined its effects in the activity of class 1 HDACs including HDAC1, 2, 3, and 8 in a human neural blastoma cell line, named SH-SY5Y. The activity-based data showed that the effectiveness of compound 5104434 was specific to HDAC1 but not other members and was able to promote enzymatic activity in a dose-dependent manner *in vitro* (Fig EV3).

To evaluate the therapeutic potential of activation of HDAC1 by using compound 5104434 in FTLD-TDP Tg mice, we gave 6-mon-old mice intraperitoneal (i.p.) injections of either the vehicle (phosphate-buffered saline [PBS] containing 1% dimethyl sulfoxide) or the compound 5104434 at 0.3, 6, or 30 mg/kg/day on 5 days per week for 1 month. We also gave i.p. vehicle injections to age-matched WT mice as a control. At this stage, the FTLD-TDP Tg mice exhibited impairments of cognitive function, motor coordination, balance, and grip strength (Tsai *et al*, 2010). After 1 month of compound 5104434 treatments, the FTLD-TDP Tg mice that received at least 6 mg/kg/day exhibited restorations of cognitive function, motor function, and the enzymatic activity of nuclear HDAC1 in the brain (Fig EV4). Notably, these treatments affected neither cell viability, nor hepatotoxic, renotoxic and body weight of mice (Fig EV5). The 6 and 30 mg/kg/day doses were similarly effective, so we chose 6 mg/kg/day as the therapeutic dose for further long-term evaluation.

Administrating 5104434 compound at 6 mg/kg/day for 2 months (Fig 6A) further shortened the escape latency of FTLD-TDP Tg mice in the Morris water maze task (Fig 6B), improved memory retention in the probe test at 24 h after escape training (Fig 6C and D), restored recognition memory in the novel object recognition test (Fig 6E), and restored motor function in the rotarod test (Fig 6F). Remarkably, although nuclear HDAC1 protein levels were unaltered following compound 5104434 treatment (Fig 7C), this treatment provided significant recovery of nuclear HDAC1 activity (Fig 7A)

---

**Figure 5.  TDP-43 interacts with HDAC1 and traps HDAC1 in inclusion bodies.**

A   Left panel: Flag-tagged full-length HDAC1 was overexpressed with myc-tagged TDP-43 in HEK-293T cells; the cell lysates were immunoprecipitated for flag and immunoblotted for TDP-43 and flag. Right panel: myc-tagged TDP-43 was overexpressed with flag-tagged full-length HDAC1 in HEK-293T cells; the cell lysates were immunoprecipitated for myc and immunoblotted for flag and TDP-43.

B   Upper left: Flag-tagged full-length HDAC1 (b.I) or various truncation mutations (b.II-IV) were overexpressed with myc-tagged TDP-43; the catalytic domain is indicated in red. Lower panel: the Western blotting of cell lysates immunoprecipitated for flag and immunoblotted for TDP-43.

C   Upper panel: Immunoprecipitation of cytosolic HDAC1 and immunoblotting of HDAC1 and TDP-43 in WT and FTLD-TDP Tg mice. Lower histogram: Quantification of immunoprecipitation results of HDAC1 and TDP-43 in WT and Tg mice. *N* = 5 mice per group, data are presented as mean $\pm$ SEM (%), *$P$ = 0.0149, ***$P$ = 0.0003 by *t*-test.

D   Western blot of HDAC1 and TDP-43 in urea-soluble fractions. *N* = 5 mice per group.

Source data are available online for this figure.

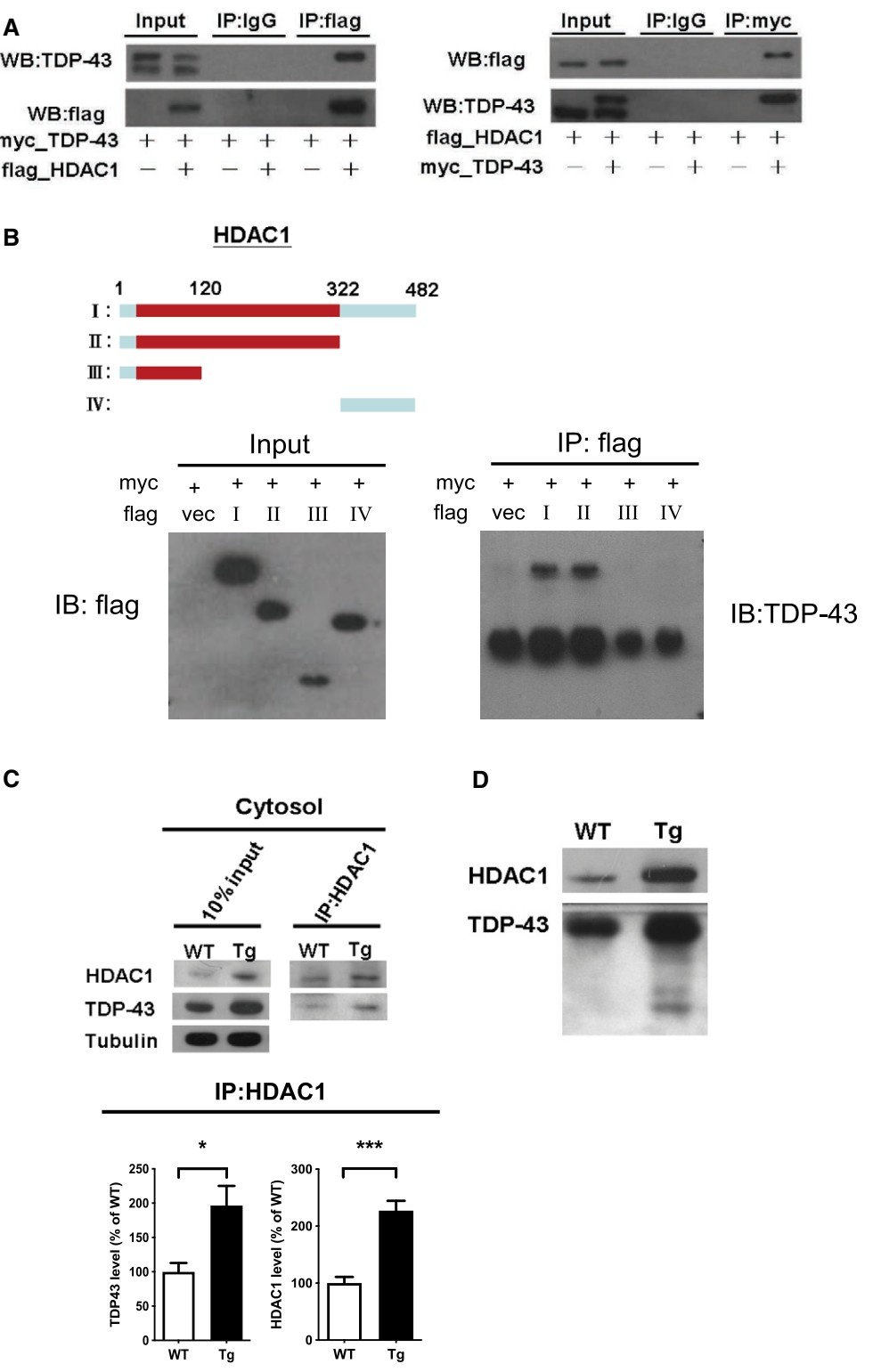

Figure 5.

and downregulation of its substrate level, e.g., acetyl-histone H3 (Lys 9/14; Fig 7B), its targets P21, and E2F1 (Fig 7D) in FTLD-TDP Tg mice. Moreover, the cell cycle aberrance, DNA damage, neuronal loss, and gliosis were ameliorated after 2 months of 5104434 treatment (Fig 7D–G). These suggest a pathogenic feature for HDAC1 function loss in aberrant cell cycle activity and DNA damage and indicate that recovery of nuclear HDAC1 activity is sufficient to restore the pathogenesis of TDP-43 proteinopathies.

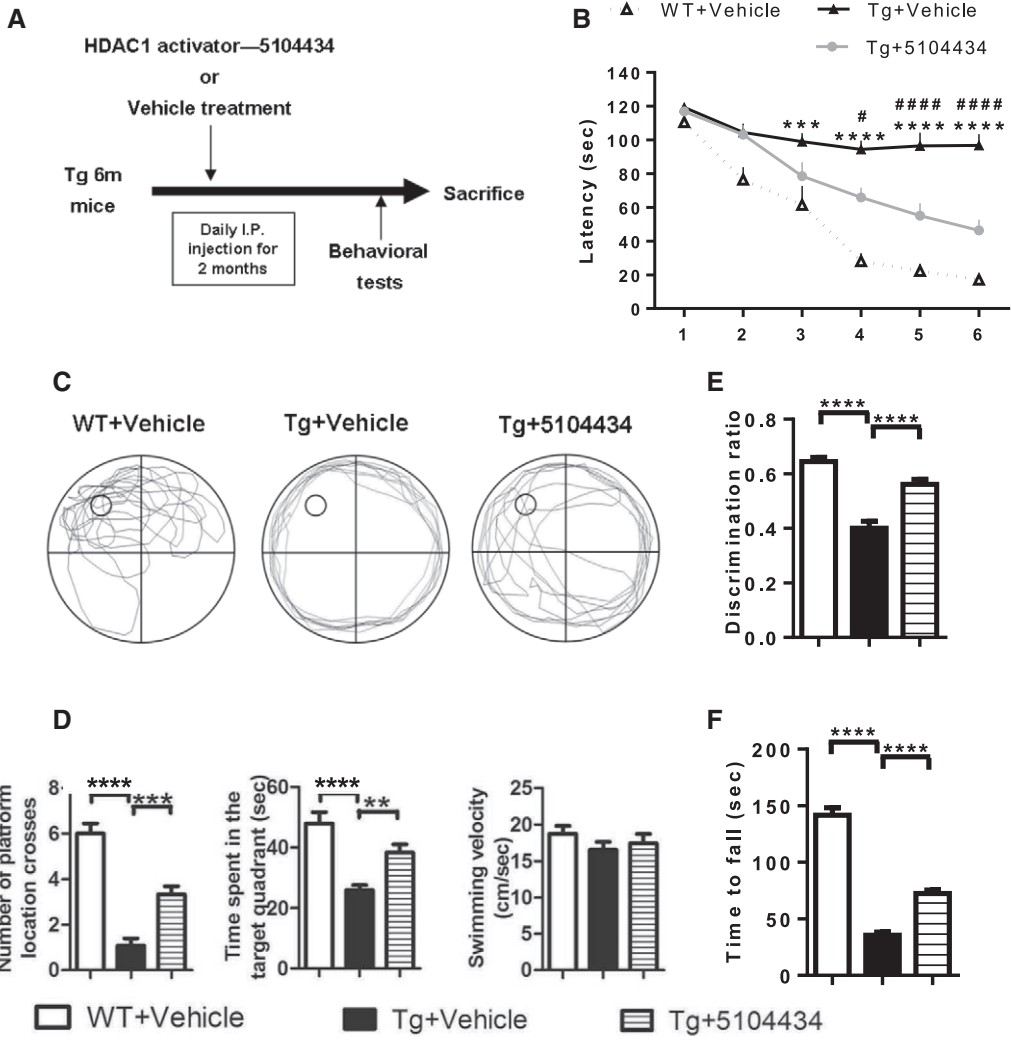

**Figure 6.  HDAC1 activator ameliorates cognitive and motor deficits in FTLD-TDP Tg mice. HDAC1 activator ameliorates cognitive and motor deficits in FTLD-TDP Tg mice.**

A   Protocols of HDAC1 activator treatment and behavioral tests. WT or FTLD-TDP Tg mice (6-mon-old) were i.p. injected with HDAC1 activator 5104434 for 2 months (6 mg/kg/day), following which they were subjected to a series of behavioral tests and then sacrificed for further examination.

B   Escape latency of mice in Morris water maze task. *WT+Vehicle mice versus Tg+Vehicle mice, #Tg+5104434 mice versus Tg+Vehicle mice. Statistical analysis by two-way ANOVA with Bonferroni's multiple comparisons. #$P < 0.05$, ***$P \leq 0.001$ and ****/####$P \leq 0.0001$, exact $P$ values are shown in Appendix Table S1.

C   The representative searching path of mice in the probe test.

D   Scores of mice with respect to the number of times they crossed the hidden platform, time spent searching in the target quadrant, and the velocity of swimming in the probe test at 24 h after escape training. **$P = 0.009$, ***$P = 0.0002$ by multiple comparison.

E   Scores of the discrimination index in the novel object recognition test.

F   Scores of mice in the rotarod test.

Data information: $N = 12$ mice per group for (B, D-F). All data are presented as mean $\pm$ SEM, ****$P < 0.0001$ by multiple comparison.

## Aberrant cell cycle activity, DNA damage, and HDAC1 deregulation are involved in the pathogenesis of FTLD-TDP in humans

To further characterize the role of TDP-43 proteinopathies in HDAC1 deregulation and its relevance to FTLD-TDP, we examined the aberrant cell cycle activity, DNA damage, and HDAC1 deregulation in the frontal cortex sections from healthy control individuals and patients with FTLD-TDP. The FTLD diagnosis was based on the participant's clinical signs and post-mortem tissue pathology, and if

TDP-43-positive inclusion bodies were identified in the brain sample, then the case was further recorded as exhibiting FTLD-TDP. With IF staining of HDAC1 and TDP-43 (Fig 8A), we conducted Sudan staining before primary antibody hybridization to prevent auto-fluorescent of lipofuscin in human tissues. We found that the frontal cortex samples from FTLD-TDP patients had significantly more cells with TDP-43 proteinopathies and HDAC1 mislocalization than the frontal cortex samples from healthy individuals did (Fig 8B) and that the two immunoreactivities were strongly correlated (Fig 8C). This suggests that TDP-43 proteinopathies play an

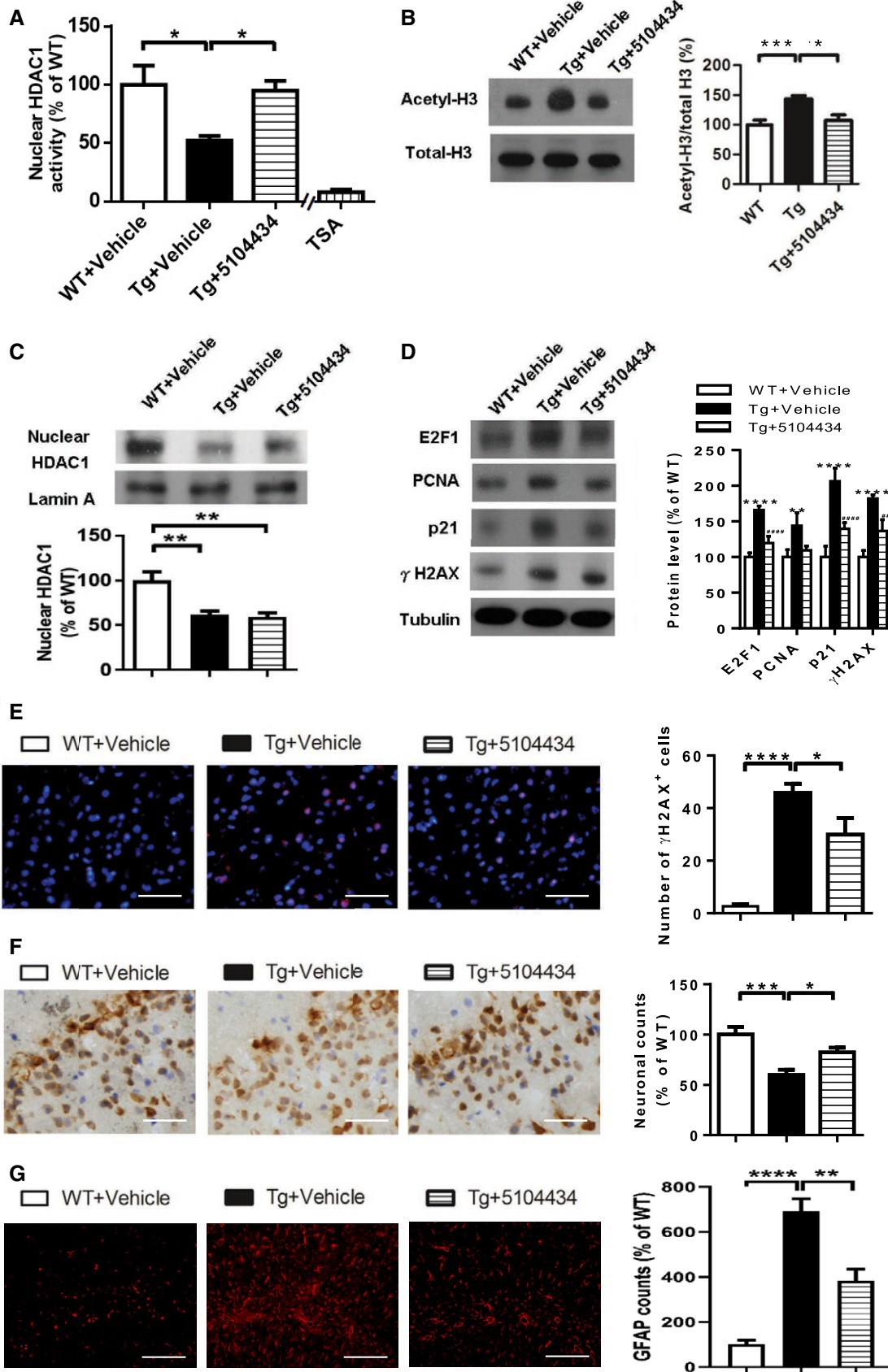

Figure 7.

**Figure 7. Pharmaceutical gain of HDAC1 activity repairs aberrant cell cycle activity, DNA damage, and neuronal loss.**

A   Nuclear HDAC1 activity in mice after 2 months of compound 5104434 treatment. The nuclear extracts of the frontal cortices and hippocampus from 8 months old of mice were separated from total tissue lysates, were immunoprecipitated for HDAC1, and were then conducted to HDAC1 activity assay. $N$ = 6 mice per group. TSA: nuclear extracts that were treated with Trichostatin A (TSA, an HDAC inhibitor) as a negative control for HDAC1-transferred fluorescent activity during the HDAC1 activity assay. *$P$ = 0.0203 (WT versus Tg-Veh) or 0.0408 (Tg-Veh versus Tg+5104434) by multiple comparison.

B   Representative data of Western blot for nuclear acetylated-histone 3 (Lys 9/14), total histone H3, and quantification of protein levels. $N$ = 6 mice per group. *$P$ = 0.0105 or ***$P$ = 0.0008 by multiple comparison.

C   Representative data of Western blot for nuclear HDAC1 levels and semi-quantification of the expression levels. $N$ = 6 mice per group. **$P$ = 0.0022 (Tg)/0.0012 (Tg+5104434) by multiple comparison.

D   Left panel: representative data of Western blot for cell cycle-related and DNA damage-related proteins. Right histogram: semi-quantification of protein levels. $N$ = 6 per group. *WT+Vehicle mice versus Tg+Vehicle mice, #Tg+5104434 mice versus Tg+Vehicle mice. Statistical analysis by multiple $t$-test with FDR correction, Q = 1%. **$P \leq 0.01$; ###$P \leq 0.001$ and ****/####$P \leq 0.0001$. Exact $P$ values are shown in Appendix Table S1.

E   IF staining of γH2AX in the frontal cortices and quantification of the immunoreactive cells from 8 months old of WT+Vehicle, Tg+Vehicle, Tg+5104434. $n$ = 9 sections per mouse, $N$ = 6 mice per group. Scale bar: 50 μm. *$P$ = 0.0421 or ****$P < 0.0001$ by multiple comparison.

F   Immunohistochemistry of NeuN and the quantification of the immunoreactive cells in the hippocampus of three groups of mice, 8-mon-old WT+Vehicle, Tg+Vehicle, and Tg+5104434. $n$ = 9 sections per mouse, $N$ = 6 mice per group. Scale bar: 50 μm.*$P$ = 0.0414 or ***$P$ = 0.0006 by multiple comparison.

G   IF staining of GFAP and the quantification of the immunoreactive cells in the hippocampus of three groups of mice, 12-month-old WT+Vehicle, Tg+Vehicle, and Tg+5104434. $n$ = 9 sections per mouse, $N$ = 6 mice per group. Scale bar: 50 μm. **$P$ = 0.002 and ****$P < 0.0001$ by multiple comparison.

Data information: All data are presented as mean ± SEM (%) except for (E, G) where number of cells in each view is presented.
Source data are available online for this figure.

essential role in HDAC1 deregulation. Furthermore, we found that the Ki67-positive and γH2AX-positive areas were highly co-localized (Fig 8D and E) and correlated (Fig 8F), suggesting that the cells with aberrant cell cycle activity were susceptible to DNA damage and shows that both aberrant cell cycle activity and DNA damage are involved in the pathogenesis of FTLD-TDP in humans. Finally, to characterize the interaction between HDAC1 and TDP-43 proteinopathies in FTLD-TDP patients, we determined the levels of HDAC1 and TDP-43 in the urea-soluble fractions from the frontal cortices by immunoblotting (Fig 8G). The quantified results showed that HDAC1 levels were obviously increased in the TDP-43-positive urea-soluble fractions (Fig 8H), which are consistent with our observations in the FTLD-TDP Tg mice. This confirms that HDAC1 deregulation is involved in the pathogenesis of FTLD-TDP in humans.

## Discussion

In this study, we used an FTLD-TDP mouse model (Tsai *et al*, 2010) to investigate the pathological mechanisms underlying TDP-43 proteinopathies. This model emulated the pathological phenotype of FTLD-TDP with progressive accumulation of cytosolic inclusion bodies and the degeneration of cognitive and motor functions. Using this model, we showed that aberrant cell cycle activity and DNA damage are observed during the pathogenesis of FTLD-TDP and further determined that HDAC1 deregulation is essential to TDP-43-mediated neurodegeneration. TDP-43 proteinopathies affected HDAC1 through protein–protein interactions, thereby reducing the levels and activity of nuclear HDAC1. In contrast, pharmacologically increasing HDAC1 activity significantly ameliorated the cognitive and motor functions impairments of FTLD-TDP Tg mice. Importantly, HDAC1 deregulation-associated aberrant cell cycle activity and DNA damage were also identified in the frontal cortex tissues from patients with FTLD. Collectively, these findings outline a pathway in TDP-43 proteinopathies by which HDAC1 deregulation causes dysfunctional transcriptional repression of cell cycle-related genes and an increased susceptibility to DNA damage. Therefore, HDAC1 activity may be a viable therapeutic target for treating FTLD and ALS.

Accumulating evidence indicates that HDAC1 plays an indispensable role in both individual development and certain diseases. In the case of individual development, global HDAC1 loss leads to early lethality (Lagger *et al*, 2002), and HDAC1 function loss interferes with neuronal differentiation (Montgomery *et al*, 2009). Furthermore, HDAC1 serves a protective role in neurons (Fischer *et al*, 2010), and HDAC1 function loss has been identified in CDK/p25-induced neurodegenerative conditions including Alzheimer's disease and stroke. Accordingly, increased HDAC1 function is associated with neuroprotective effects against ischemia-induced neuronal death (Kim *et al*, 2008) and Huntington's disease in a *Caenorhabditis elegans* model (Bates *et al*, 2006). Our findings indicate a previously unknown mechanism whereby HDAC1 deregulation in TDP-43 proteinopathies leads to DNA damage and substantial cell cycle deregulation, which results in neuronal loss and brain degeneration. Therefore, in addition to the neurotoxicity of cytosolic TDP-43 inclusions, we extend the role of HDAC1 deregulation to the pathogenesis of FTLD-TDP and ALS. Notably, there are currently no approved drugs that specifically target HDAC1, but we herein provide evidence that HDAC1 could be a therapeutic target for FTLD and ALS.

Under physiological conditions, cell cycle activity in post-mitotic neurons is normally suppressed. HDAC1 is a transcriptional repressor for many cell cycle-related genes (Hassig *et al*, 1998), so its deregulation leads to cell cycle dysregulation in numerous neurodegenerative disorders (Kim *et al*, 2008). Abnormal activity of cell cycle-related genes can induce neuronal apoptosis, which aggravates pathogenesis (Ranganathan & Bowser, 2003). Accordingly, HDAC1 deregulation is a potential causal factor in the progression of neurodegeneration. In this study, we showed that disease onset in FTLD-TDP Tg mice occurred in parallel with reduced nuclear HDAC1 levels and activity, which caused deregulated transcriptional repression in the epigenetic modulation of cell cycle-related genes. In addition, we found that acetyl-histone 3 was upregulated in our FTLD-TDP mouse model, which further implies that HDAC1 deficits induce neurodegeneration (Narayan *et al*, 2015). Therefore, our model suggests that stable HDAC1 is essential for repressing cell cycle-related genes and maintaining neuronal homeostasis.

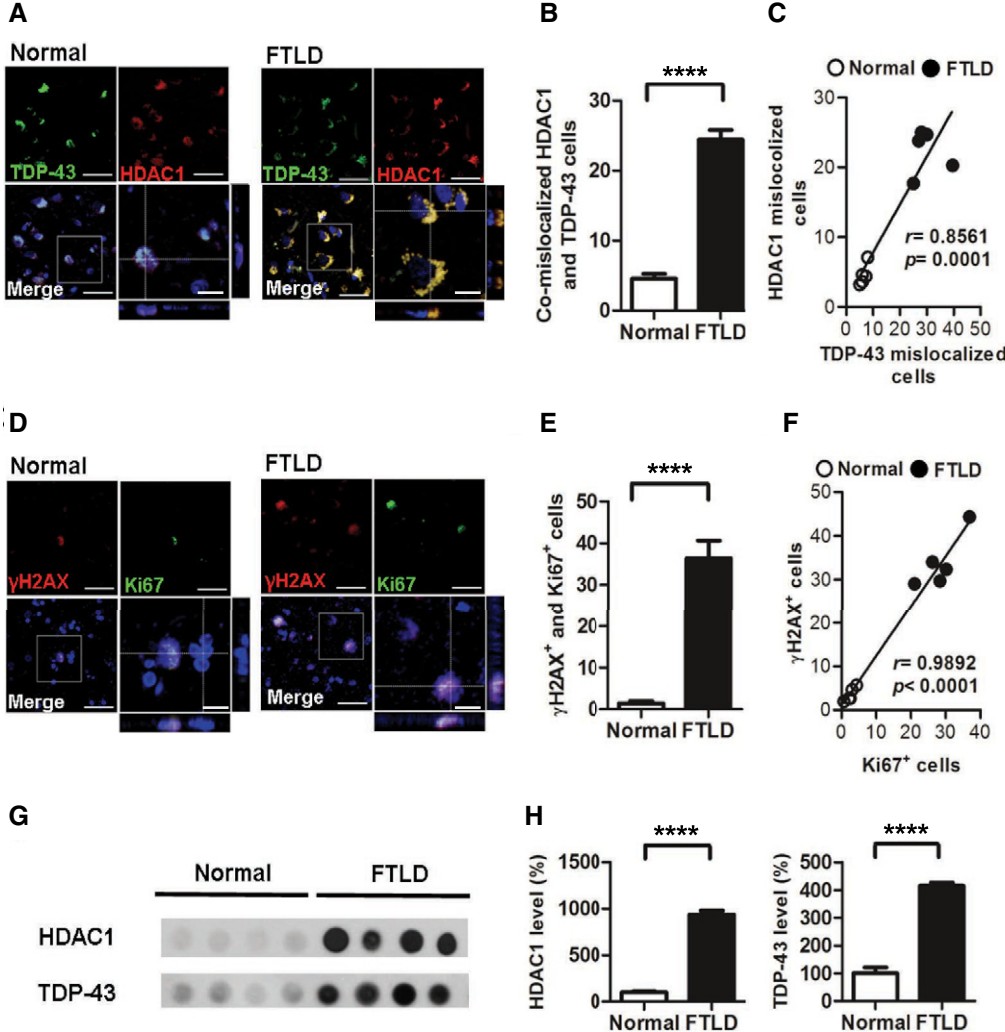

**Figure 8. Deregulation of HDAC1 is involved in aberrant cell cycle activity and DNA damage in the frontal cortices from patients with FTLD-TDP.**

A   Representative IF staining of TDP-43 and HDAC1 in the frontal cortices from normal individuals and patients with FTLD-TDP. Scale bar: 50 μm. The circled area is emphasized for showing the distribution of immunoreactivity in cell subregions. Scale bar: 15 μm.

B   Quantification of cells with co-mislocalized HDAC1 and TDP-43 from each view of microscope. *N* = 5 per group.

C   Linear regression analysis of cells with HDAC1 mislocalization and TDP-43 proteinopathies. Total cell counts: 3,000 per samples. *P* = 0.0001 by Pearson correlation analysis.

D   Representative IF staining of γH2AX and Ki67 in the frontal cortices from normal individuals and patients with FTLD-TDP from each view of microscope. Scale bar: 50 μm. The circled area is emphasized for showing the distribution of immunoreactivity in cell subregions. Scale bar: 15 μm.

E   Quantification of cells with γH2AX and Ki67 immunoreactivity. *N* = 5 per group.

F   Linear regression analysis of cells with DNA damage and aberrant cell cycle activity. Total cell counts: 3,000 per samples. *P* < 0.0001 by Pearson correlation analysis.

G   Dot blot of HDAC1 and TDP-43 in urea-soluble fractions from the frontal cortex of normal individuals and patients with FTLD-TDP.

H   Quantification of HDAC1 and TDP-43 expression levels in urea-soluble fractions. *N* = 5 per group.

Data information: All data represent mean ± SEM, ****$P$ < 0.0001 by *t*-test.
Source data are available online for this figure.

We further observed that DNA double-strand breaks were involved in the pathogenesis of TDP-43 proteinopathies. Loss of HDAC1 function may play an essential modulatory role in these breaks by inducing a hypersensitization of DNA, which may in turn increase the susceptibility of DNA to damage (Cerna *et al*, 2006). Furthermore, HDAC1 was identified as participating in the DNA double-strand break response and DNA repair by interacting with the fused-in-sarcoma ribonucleoprotein (Wang *et al*, 2013). HDAC1 also contributes to the maintenance of genomic stability with SIRT1 and protein kinase ataxia–telangiectasia mutated in neurons (Dobbin *et al*, 2013). Therefore, our findings suggest that in the pathogenesis of FTLD-TDP, HDAC1 deregulation, in the form of deficiency or dysfunction, leads to a deficiency of double-strand break responses and DNA repair and thus facilitates pathogenesis.

Under physiological conditions, TDP-43 is a nuclear DNA/RNA-binding protein involved in transcriptional repression and mRNA

processing. TDP-43-induced ubiquitin-positive cytoplasmic inclusions are the pathological hallmarks of FTLD-TDP and ALS (Wang *et al*, 2008). However, the structural properties of aggregated TDP-43 and its relationship with pathogenesis remain unclear. Previous studies have shown the propensity of TDP-43 to form amyloid oligomers, which damage neurons by neurotoxicity in the brains of both FTLD-TDP mice and patients (Fang *et al*, 2014). Furthermore, the level of lysine acetylation of TDP-43 was reported as a novel post-translational modification controlling TDP-43 function and aggregation (Cohen *et al*, 2015), and cytosolic HDAC6 is involved in this critical physiological/pathological switch. Here, our findings indicate that HDAC1-TDP-43 interactions result in the export of nuclear HDAC1, which may in turn promote TDP-43 acetylation and aggravate TDP-43 proteinopathies. In addition, we determined that HDAC1 structurally interacts with TDP-43 via its catalytic domain. Our findings show that HDAC1 and TDP-43 are progressively exported from the nucleus to the cytosol during pathogenesis (Fig 4) and that HDAC1 interacts with soluble TDP-43 in both whole cell lysates *in vitro* and cytosolic fractions from the brains of FTLD-TDP mice (Fig 5B). Importantly, we found that HDAC1 is trapped in TDP-43-positive inclusions (Figs 5C and 8) in samples from FTLD-TDP mice and patients. Together, these findings indicate that mislocalized TDP-43 is the leading cause of HDAC1 mislocalization, that HDAC1 is trapped in TDP-43 aggregates and inclusions, and explain the observed reduction in nuclear HDAC1 levels, as well as no mislocalized TDP-43 and HDAC1 can be detected at the early time of disease onset (2 months old of Tg mice, Tsai *et al*, 2010), indicating that HDAC1 dysregulation represents a second pathogenic wave contributing to cognitive deficits. Moreover, TDP-43 proteinopathies may also influence HDAC1 activity due to the interference of the Zn-dependent catalytic domain (Fig 5), such that HDAC1's functional domain is blocked.

Both TDP-43 and HDAC1 can shuttle between the nucleus and cytosol, but they are primarily located in the nucleus and rarely expressed in the cytosol (Kim *et al*, 2010). Whether HDAC1 has a cytosolic function in the brain remains unknown, but cytosolic HDAC1 is regarded as a potential mechanism for damage initiation in demyelinating conditions (Kim & Casaccia, 2010). Therefore, we speculate that damage to axon transport mechanisms may interfere with the nuclear reuptake of HDAC1 and thus reduce nuclear HDAC1 levels. Interestingly, HDAC1 binds exportin 1/CRM-1 for export from the nucleus in a calcium-dependent manner only under pathological conditions, not physiological ones. Cytosolic HDAC1 may subsequently form complexes with motor proteins and microtubules, thus causing axonal transport blockade and axon damage (Kim *et al*, 2010). Given the essential role of TDP-43 in multiple physiological functions, its role in the pathogeneses of FTLD-TDP and ALS, and that both TDP-43 and HDAC1 are involved in epigenetically induced transcriptional repression, there are several questions that are worth investigating in future studies. These include the mechanisms by which TDP-43 interacts with HDAC1 under physiological and pathological conditions, whether they leave the nucleus as a bound complex, whether they can be returned to the nucleus after nuclear export, and whether TDP-43 is involved in DNA repair.

Accumulating evidence indicates that histone modification is a promising therapeutic strategy for numerous neurodegenerative diseases (Fischer *et al*, 2010). For example, deregulation of chromatin acetylation has been identified in aged mice, and HDAC inhibition can reverse aging-related cognitive decline in mice (Peleg *et al*, 2010). However, despite considerable evidence supporting the neuroprotective role of HDACs, the therapeutic approaches that specifically target HDAC1 remain poorly defined. For example, HDAC inhibitors have shown efficacy in preventing neuronal toxicity in animal models of Alzheimer's disease, ALS, and Huntington's disease (Coppede, 2014), but each kind of HDAC has its own associated genes and promotes different functions. Therefore, manipulating specific HDACs should be the preferred strategy for developing novel therapeutics. For example, specifically promoting HDAC1 activity with pharmacological SIRT1 activators or by increasing HDAC1 function via genetic overexpression can reduce DNA damage in mouse models of Alzheimer's disease and ischemia (Kim *et al*, 2008; Dobbin *et al*, 2013). In this study, we characterized HDAC1 function loss in FTLD-TDP mice and found that pharmacologically increasing HDAC1 activity can compensate for nuclear HDAC1 deficiencies and thereby promote the recovery of cognitive and motor functions. We have also performed the nuclear HDAC2 activity assay. Despite that HDAC1 and HDAC2 share highly conserved N-terminal part including the protein–protein interaction domain (1–52) and the catalytic domain (12–322), there is no difference in the HDAC2 activity between WT and Tg mice (Unpublished observations, Pei-Chuan Ho). Therefore, it is more convincing that only HDAC1, but not HDAC2, is affected by nuclear TDP-43 mislocalization, thus contributes to TDP-43 proteinopathies. Because effective therapies for FTLD-TDP and ALS are urgently needed, we herein provide mechanistic insights into HDAC1 deregulation in TDP-43 proteinopathies and outline a promising therapeutic strategy for treating FTLD-TDP and ALS by targeting HDAC1.

## Materials and Methods

### Wild-type mice and FTLD-TDP Tg mouse model

Experimental procedures for handling the mice were in accordance with the guidelines of the Institutional Animal Care and Use Committee (IACUC) of NCKU. All animal experimental protocols were approved by IACUC of NCKU. The mice were housed no more than 5 per cage in a pathogen-free room maintained on a 12-h light–dark cycle, fed with standard rodent chow, and given sterilized drinking water *ad libitum*. The frontotemporal lobar degeneration with ubiquitin ($^+$) inclusions (FTLD-TDP) mouse model carried a transgenic full-length mouse TDP-43 cDNA, under the transcriptional control of an 8.5-kb promoter region of the CaMKII gene (Tsai *et al*, 2010). The assays of behavioral performances and pathological features of the homozygous ($^{+/+}$) FTLD-TDP Tg male mice were performed as previously described (Tsai *et al*, 2010). Wild-type FVB male mice with 2 months old of age were purchased from the animal center of the NCKU.

### Cell models

HEK-293T and SH-SY5Y cell lines were maintained as a monolayer in DMEM/F12 medium (Thermo, catalog NO. 11320033)

supplemented with 10% fetal bovine serum (Bioindustry, 04-001-1A-US) and were kept at low passage and regularly tested with mycoplasma incase contamination.

## Immunostaining

To prepare the tissue for immunofluorescence staining, adult mice were anesthetized and perfused transcardially with PBS and 4% paraformaldehyde (PFA). The brain of each mouse was then removed and immersed in 4% PFA solution for 2 h and then dehydrated by a gradient of 15, 20, and 30% sucrose solutions. Cryosections were prepared by slicing the tissue to 12 μm thickness, which was followed by blocking with 5% normal donkey serum and incubation with the following primary antibodies: TDP-43 (Proteintech, 10782-2-AP; Rosemont, IL, 1:300), HDAC1 (Thermo, PA1-860; Waltham, MA, 1:300), Ki67 (Abcam, ab15580; Cambridge, UK, 1:750), γH2AX (Millipore, 05-636; Billerica, Ma, 1:300), SMI32 (Covance, SMI-32R; Prinston, NJ, 1:100), GFAP (Millipore, AB5804, 1:300), and NeuN (Millipore, MAB377, 1:300). Alexa Fluor-conjugated secondary antibodies (Thermo, 1:300) were used to detect the primary antibodies. Finally, the sections were incubated with DAPI and mounted on coverslips with mounting medium (Dako). The value of each mouse was acquired from five random views of the cortex or the hippocampus in one section; total nine sections within dorsal to ventral cortex or hippocampus were evaluated per mouse. To prevent auto-fluorescence in human tissues, the sample from human patients was conducted with Sudan staining following previous described procedures before the primary antibody hybridization (Hendrickson *et al*, 2011). All sections were observed and photographed using a laser scanning confocal microscope (Nikon C1-Si). The immunoreactive cells were quantified in a blinded manner; each human sample was randomly counted over 20 image views under 20X objective lens from the regions frontal cortices; each experimental mouse was also randomly counted over 20 image views under 20× objective lens from coronal sections in the regions of FrA and motor cortex M1/M2 in the frontal cortices, the regions were related to FTLD pathogenesis (Tsai *et al*, 2010; Wils *et al*, 2010).

## RT–PCR

Total RNA was isolated using TRIzol reagent (Thermo), according to manufacturer's instructions. Reverse transcription of equal amounts of total RNA was performed using the Superscript II First-Strand Synthesis Kit (Thermo), according to manufacturer's instructions. Semi-quantitative conditions were obtained for the following sets of primers: 5′ TGTCCAATCCTGGT GATGTCC (sense) and 5′ TCAGA CACCAGAGTGCAAGAC (antisense) for p21; 5′ TGATGAAGGCCCT TAAGTGG (sense) and 5′ GGCCACTTGGACATAGACAT (antisense) for Cyclin E; 5′ TAGCCCTGGGAAGACCTCAT (sense) and 5′ CCCCAAAGTCACAGTCAAAGAG (antisense) for E2F1. 5′ACAACT CCGCCACCATGTTTG (sense) and 5′GCCTAAGATGCTTCCTCATC TTC (antisense) for PCNA. 5′GACCCCTTCATTGACCTCAAC (sense) and 5′TCTTACTCCTTGGAGGCCATG (antisense) for GAPDH. PCR cycle conditions were as follows: 94°C for 45 s, 56°C for 45 s, and 72°C for 1 min. In total, 29 cycles were used for PCR and quantification of all genes. Band quantifications were normalized to those obtained for GAPDH for each animal.

## Western blotting

For analysis of soluble proteins, cells or tissues were homogenized in RIPA lysis buffer, and the total protein content was extracted as detailed in a previously described protocol (Wang *et al*, 2012). The extracts were analyzed by 8–12% sodium dodecyl sulfate–polyacrylamide gel electrophoresis, followed by blotting with the following antibodies: E2F1 (Abcam, ab137415, 1:1,000), Cyclin A (Santa Cruz, H-432, 1:2,000), PCNA (Abcam, ab15497, 1:1,000), P21 (Abcam, ab7960, 1:2,000), TDP-43 (Proteintech, 10782-2-AP; Rosemont, IL, 1:1,000), HDAC1 (Thermo, PA1-860; Waltham, MA, 1:1,000), γH2AX (Millipore, 05-636, 1:800), acetyl-histone H3 (Santa Cruz, SC-8655, 1:500), total histone H3 (Cell Signaling, #4499, 1:1,000), and tubulin (Abcam, ab4074, 1:5,000). After primary antibody binding, the blots were incubated at room temperature with the appropriate secondary antibodies and Western Lightning Plus-ECL (PerkinElmer; Waltham, MA). All cell data are repeated at least 3 independent experiments. For quantitative analysis, relative intensities of the bands were normalized against those of internal controls and expressed as mean ± SEM.

## Nuclear HDACs activity assay

For *in vivo* HDAC1 activity detection, the nuclear protein fractions were isolated with a nuclear extraction kit (Active Motif; Carlsbad, CA) following the manufacturer's directions. Briefly, the dissected brain samples were homogenized in hypotonic buffer for 15 min on ice and then centrifuged at 850 *g* for 10 min. The pellet was resuspended in hypotonic buffer with 0.5% detergent for 15 min on ice and centrifuged at 14,000 *g* for 1 min, and the complete lysis buffer was added to dissolve the nuclear protein. Total 800 μg of nuclear protein from each sample was subjected to IP for nuclear HDAC1. The IP products were used to detect HDAC1 activity by using an activity assay kit (Enzo Life Sciences, BML-AK500-0001; Farmingdale, NY) following the manufacturer's instructions. The sample treated with Trichostatin A (TSA), an HDAC inhibitor, was used as negative control to abrogate HDAC1-transferred fluorescent activity (Kim *et al*, 2008).

## Urea fraction extracts

For analysis of insoluble proteins, tissues were dissected and sequentially extracted with buffers of increasing strengths, as previously described (Fang *et al*, 2014). In brief, the forebrains were extracted sequentially at 5 ml/g with low-salt (LS) buffer (10 mM Tris, pH 7.5, 5 mM EDTA, 1 mM DTT, 10% sucrose, and a mixture of protease inhibitors), high-salt Triton X-100 (TX) buffer (LS + 1% Triton X-100 + 0.5 M NaCl), myelin flotation buffer (TX buffer containing 30% sucrose), and Sarkosyl (SARK) buffer (LS + 1% N-lauroylsarcosine + 0.5 M NaCl). The SARK-insoluble materials were further extracted in 0.25 ml/g urea buffer (7 M urea, 2 M thiourea, 4% 3-[(3-Cholamidopropyl) dimethylammonio]-1-propanesulfonate, 30 mM Tris, pH 8.5). The urea-soluble proteins were then analyzed by Western blotting.

## Comet assay

CometAssay® kit (Trevigen; Gaithersburg, MD) was used for detection of DNA damage, according to the manufacturer's instructions.

Briefly, the dissected mice forebrains were immersed into 2 ml of ice-cold PBS containing 20 mM EDTA, minced into small pieces by using scissors, and homogenized into single cells. The cells obtained were resuspended at a density of $1 \times 10^5$ cells/ml, embedded in LM Agarose (1:10), and placed on the CometSlide™. The slide was treated with lysis solution, alkali, and subjected to alkaline electrophoresis. Finally, samples were stained with intercalating dye and visualized by fluorescence microscopy.

### Drug treatment

The HDAC1 activator compound 5104434 (ChemBridge; San Diego, CA) was dissolved in DMSO with a stock concentration of 3 mg/ml, and the administration of 5104434 was conducted in 6-mon-old WT or Tg mice for 1 or 2 months, using daily i.p. injection, 5 days per week. To test the feasible therapeutic dose, three groups receiving 0.5, 6, or 30 mg/kg/day were designed. Theophylline (Sigma; Fremont, CA) was dissolved in PBS and injected daily via i.p. at 1.8 mg/kg for 2 months, and 1% DMSO in PBS or PBS alone served as the vehicle control.

### Morris water maze task and probe test

Morris water maze task was conducted as described previously (Wu *et al*, 2017). Briefly, animals were subjected to four trials per session and one session a day. A complete test consisted of six sessions over 6 days. Each mouse had 120 s to reach the hidden platform in the water, and the average time per session was recorded as escape latency. After the escape training, the platform was removed for the probe trial test, which tested the retention of spatial memory at 24 h after training. The number of times the mice crossed the probe, the time spent in the target quadrant, and swim speed were recorded and analyzed by video recording.

### Novel object recognition test

Novel object recognition test was conducted as described in previous studies (Wu *et al*, 2016). On day 1 of training, mice were subjected to a new open-field cage (35 × 45 × 40 cm) and placed individually in the cage for a 5-min familiarization trial with no objects. Subsequently, the mice went through three sessions of 5-min sample trials with two of the same objects positioned at specific places in the cage. Each session was suspended by a 3-min interval period where the mice were returned to their original cages. Twenty-four hours later, the mice were subjected to the test trial for 5 min with two objects in the cage (a familiar object and a novel object) positioned at the same specific places. The discrimination index (DI) was calculated as follows: DI = novel object exploration time/total exploration time.

### Rotarod test

For the rotarod (Harvard Apparatus; Holliston, MA) test, an accelerating mode was used with speed acceleration to 20 rpm in 2 min, and the time that the mouse could keep on the rotating rod was recorded. Two days of training course, 3 trials per day, was conducted for the mice. The average time on the third day was recorded for evaluation of motor function.

### Human samples

Post-mortem brain samples of normal individuals and patients with FTLD were kindly provided through Dr. Lee-Way Jin by the Alzheimer's Disease Center at University of California Davis, Sacramento, California. Informed consented autopsies to share research tissue after death were obtained from all patients with Institutional Review Board approval. The study was approved by the Institutional Review Board of NCKU Hospital (B-ER-103-180) based on the ethical standards prescribed by the World Medical Association (WMA) Declaration of Helsinki and the Department of Health and Human Services Belmont Report. FTLD-TDP subtypes were designated using consensus nomenclature and diagnosed following the original descriptions (Kao *et al*, 2015).

### Statistical analysis

All data are statistically analyzed and graphically represented using GraphPad Prism 8 (GraphPad Software, San Diego, CA, USA), which were normal distributed and are presented as the mean ± SEM. Mice were allocated to experimental groups on the basis of their genotype and randomized within the given group. Sample sizes were typically between $n = 5$–6 mice per group and at least $n = 4$–9 sections for IF staining per mouse. To minimize subjective bias, sample identity was blinded by giving an identification number to each mouse before experiment start. For the Morris water maze task, escape latency was analyzed by repeated measures analysis of variance (ANOVA) with Bonferroni test for multiple comparisons. Other independent experiments with multiple groups were compared with each other by one-way ANOVA followed by *post hoc* Tukey's test. Two-tailed Student's *t*-test was used to analyze two-group comparisons and Mann–Whitney test in non-normal distributed data. The D'Agostino–Pearson normality test was used, and F-test was used to compare equality of variances. Differences were considered statistically significant at $*P < 0.05$, $**P \leq 0.01$; $***P \leq 0.001$, and $****P \leq 0.0001$ as indicated.

**Expanded View** for this article is available online.

### Acknowledgements

The authors are grateful to University of California Davis Alzheimer's Disease Center, funded by National Institute on Aging (NIA, grant #P30AG10129) for collecting and providing the human samples. This study was performed, in part, with support from Taiwan Ministry of Science and Technology (NSC-102-2320-B-006-040-MY3, MOST-103-2321-B-006-028, MOST-104-2321-B-006-010, MOST-105-2321-B-006- 002, MOST-105-2628-B-006-016-MY3, and MOST-106-2628-B-006-001-MY4).

### Author contributions

I-FW: collection and assembly of data, data analysis and interpretation, and manuscript writing. L-WJ: sample collection and assembly of data and data analysis and interpretation. C-CW: collection and assembly of data, data analysis and interpretation, and manuscript writing. W-YW: collection and assembly of data and data analysis and interpretation. P-CH: collection and assembly of data and data analysis and interpretation. Y-CL: collection and assembly of data and data analysis and interpretation. K-JT: conception and design, collection and assembly of data, data analysis and interpretation, manuscript writing, and final approval of the manuscript.

**The paper explained**

**Problem**

TAR DNA-binding protein 43 (TDP-43) has been implicated in frontotemporal lobar degeneration (FTLD). Histone deacetylase 1 (HDAC1) is involved in DNA repair and neuroprotection in numerous neurodegenerative diseases. However, the pathological mechanisms of FTLD underlying TDP-43 proteinopathies are unclear.

**Results**

The results demonstrated cell cycle aberrance and DNA damage are involved in the degenerative progress of FTLD and investigated the role of HDAC1 in TDP-43 proteinopathies. This hypothesis is consistent with non-clinical and clinical findings.

**Impact**

The paper proposes a framework for the generation of new hypotheses and the conduct of additional studies. Examples include further understanding: HDAC1 deregulation is involved in the pathogenesis of TDP-43 proteinopathies, and HDAC1 is a potential target for therapeutic interventions in FTLD. By restoring, HDAC1 activity may be a feasible approach to treating FTLD.

## Conflict of interest

The authors declare that they have no conflict of interest.

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
