## [Review Process File · EMBO Molecular Medicine]

HDAC1 Dysregulation Induces Aberrant Cell Cycle and DNA Damage in Progress of TDP-43 Proteinopathies

Cheng-Chun Wu, Lee-Way Jin, I-Fang Wang, Wei-Yen Wei, Pei-Chuan Ho, Yu-Chih Liu, and Kuen-Jer Tsai

DOI: [10.15252/emmm.201910622](https://doi.org/10.15252/emmm.201910622)

Corresponding author(s): Kuen-Jer Tsai (kjtsai@mail.ncku.edu.tw)

Review Timeline:

Submission Date:	9th Apr 19
Editorial Decision:	9th May 19
Revision Received:	9th Aug 19
Editorial Decision:	10th Sep 19
Revision Received:	9th Apr 20
Accepted:	24th Apr 20

Editor: Jingyi Hou

Transaction Report:

9th May 2019

Dear Dr. Tsai,

Thank you for the submission of your manuscript to EMBO Molecular Medicine. We have now heard back from three referees who evaluated your manuscript.

As you will see from the comments below, the referees find the manuscript to be of potential interest and highlight that it provides both novelty and clinical value. However, they also raise substantial concerns about your work, which should be convincingly addressed in a major revision of your manuscript. Additional experiments and controls should be performed to strengthen the molecular mechanism as suggested by all reviewers. In particular, we would strongly encourage you to examine the cellular localization of HDAC1 in brain samples from frontotemporal lobar degeneration patients as recommended by reviewer #2 and reviewer #3 to improve the clinical relevance of the study.

We would welcome the submission of a revised version within three months for further consideration and would like to encourage you to address all the criticisms raised as suggested to improve conclusiveness and clarity. Please note that EMBO Molecular Medicine strongly supports a single round of revision and that, as acceptance or rejection of the manuscript will depend on another round of review, your responses should be as complete as possible.

I look forward to receiving your revised manuscript.

Yours sincerely,
Jingyi Hou

Jingyi Hou
Editor
EMBO Molecular Medicine

*** Instructions to submit your revised manuscript ***

** PLEASE NOTE ** As part of the EMBO Publications transparent editorial process initiative (see our Editorial at <http://embomolmed.embopress.org/content/2/9/329>), EMBO Molecular Medicine will publish online a Review Process File to accompany accepted manuscripts.

To submit your manuscript, please follow this link:

Link Not Available

- 1) a .doc formatted version of the manuscript text (including Figure legends and tables). Please make sure that the changes are highlighted to be clearly visible to referees and editors alike.
- 2) separate figure files*
- 3) supplemental information as Expanded View and/or Appendix. Please carefully check the authors guidelines for formatting Expanded view and Appendix figures and tables at <http://embomolmed.embopress.org/authorguide#expandedview>
- 4) a letter INCLUDING the reviewers' reports and your detailed responses to their comments (as Word file)

Also, and to save some time should your paper be accepted, please read below for additional information regarding some features of our research articles:

- 5) The paper explained: EMBO Molecular Medicine articles are accompanied by a summary of the articles to emphasize the major findings in the paper and their medical implications for the non-specialist reader. Please provide a draft summary of your article highlighting
 - the medical issue you are addressing,
 - the results obtained and
 - their clinical impact.

- 6) For more information: There is space at the end of each article to list relevant web links for

further consultation by our readers. Could you identify some relevant ones and provide such information as well? Some examples are patient associations, relevant databases, OMIM/proteins/genes links, author's websites, etc...

7) Author contributions: the contribution of every author must be detailed in a separate section (before the acknowledgments).

8) EMBO Molecular Medicine now requires a complete author checklist (<http://embomolmed.embopress.org/authorguide#editorial3>) to be submitted with all revised manuscripts. Please use the checklist as guideline for the sort of information we need WITHIN the manuscript as well as in the checklist. This is particularly important for animal reporting, antibody dilutions (missing) and exact p-values and n that should be indicated instead of a range.

9) Every published paper now includes a 'Synopsis' to further enhance discoverability. Synopses are displayed on the journal webpage and are freely accessible to all readers. They include a short stand first (maximum of 300 characters, including space) as well as 2-5 one sentence bullet points that summarise the paper. Please write the bullet points to summarise the key NEW findings. They should be designed to be complementary to the abstract - i.e. not repeat the same text. We encourage inclusion of key acronyms and quantitative information (maximum of 30 words / bullet point). Please use the passive voice. Please attach these in a separate file or send them by email, we will incorporate them accordingly.

You are also welcome to suggest a striking image or visual abstract to illustrate your article. If you do please provide a jpeg file 550 px-wide x 400-px high.

10) A Conflict of Interest statement should be provided in the main text

11) Please note that we now mandate that all corresponding authors list an ORCID digital identifier. This takes <90 seconds to complete. We encourage all authors to supply an ORCID identifier, which will be linked to their name for unambiguous name identification.

Currently, our records indicate that there is no ORCID associated with your account.

Please click the link below to provide an ORCID:

Link Not Available

12) The system will prompt you to fill in your funding and payment information. This will allow Wiley to send you a quote for the article processing charge (APC) in case of acceptance. This quote takes into account any reduction or fee waivers that you may be eligible for. Authors do not need to pay any fees before their manuscript is accepted and transferred to our publisher.

Photos 400-800 DPI

*Additional important information regarding figures and illustrations can be found at <http://embomolmed.embopress.org/authorguide#figures>

***** Reviewer's comments *****

Referee #1 (Remarks for Author):

The manuscript by Wu and colleagues by Wu et al. describes the involvement of HDAC1 in a model of TDP-43-linked FTD. In a previously published mouse overexpressing TDP-43 in forebrain neurons, they find that TDP-43 mislocalization to the cytosol is associated with increased markers of cell cycle activity, DNA ds-breaks, and decreased nuclear localization and activity of HDAC1. They also show, using both in cellulo and in vivo systems, that TDP-43 interacts with HDAC1. Hence, they propose that TDP-43 mislocalization to the cytosol traps HDAC1, causing its redistribution from the nucleus to the cytosol. As a result, HDAC1 in TDP-43 mice fails to regulate cell cycle and aid in DNA repair. They surmise that deficiency of HDAC1 activity in the nucleus participates in disease pathogenesis and to overcome the defect they treat mice with a specific HDAC1 activator. The treatment results in an improvement of the disease phenotype (memory and motor performance) of the mouse and a reduction of the DNA damage and cell cycle markers. Lastly, they show that HDAC1 and the other markers altered in the mouse model are similarly altered in autoptic brain tissue from patients affected by FTD with TDP-43 pathology. Overall, the findings are interesting and nicely support the involvement of HDAC1 in TDP-43 FTD. HDAC1 is proposed to be among the many proteins that become mislocalized in the cytosol secondarily to TDP-43 accumulation. Nevertheless, the manuscript would benefit from revisions of interpretation of the results and some additional data.

- Throughout the manuscript, the authors state that certain observed alterations (i.e., DNA damage, cell cycle markers, etc.) must play a role in the pathogenesis of the disease. These statements must be thoroughly corrected. The only alteration that they can claim to play a disease role is the decline in HDAC1 activity, if the effects of compound 5104434 are specifically HDAC1 activation. The other changes downstream of HDAC1 may or may not directly participate in disease pathogenesis. They may also be concurring events, because no specific interventions address them individually, in relationship to disease phenotype.

- Major point: it appears that the only reference to compound 5104434 is an international patent held by the corresponding author since 2010 (WO2010011318). The effects of 5104434 on the mouse are the clincher of this work, because they link together the other observations. More information and data on this compound (potency, specificity, bioavailability, safety, etc.) must be provided, and potential conflicts of interests must be clearly stated.

- A genetic approach to confirm the requirement of HDAC1 for the described effects of compound 5104434 would be reassuring.

- As expected, compound 5104434 does not modify the localization of HDAC1 or TDP-43

mislocalization, but presumably increases the activity of residual nuclear HDAC1. However, it is necessary to define whether it affects neuronal pathology, not just in terms of markers of DNA damage and cell cycle, but also of neuroinflammation, gliosis, etc.

- The co-IP experiments in figure 5 must be quantified, especially 5C.

Referee #2 (Comments on Novelty/Model System for Author):

Human sample details need to be described more in details

Referee #2 (Remarks for Author):

Frontotemporal lobar degeneration (FTLD) and amyotrophic lateral sclerosis (ALS) are among the most common neurodegenerative diseases, yet there are still no effective cures. In this manuscript, Wu et al. used a FTLD-TDP Tg mouse model driven by CamKII α promoter (generated by the authors' lab previously) that overexpresses TDP43 in frontal cortex to dissect the potential mechanisms and to test potential therapies. From their previous study, the mice manifest mislocalisation of TDP43 in neurons and develop early symptoms of impaired learning/memory, and subsequent hippocampal atrophy, as well as motor dysfunction. Here, they further reported the coexistence of abnormal activation of cell cycle pathway and DNA damage in neurons with TDP43 proteinopathy (mislocalisation). Interestingly, the nuclear protein HDAC1 was also found transported abnormally to cytoplasm in neurons, and seemed to be concomitant with TDP43 mislocalisation. These findings were manifested in certain degrees in brain samples of human FTLD-TDP patients. Further biochemical experiments revealed that TDP43 directly binds to the catalytic domain of HDAC1, therefore might contributing to the mislocalisation of HDAC1 as a consequence of TDP43 export to the nucleus. Finally, upon treating FTLD-TDP Tg mice with a HDAC1 specific inhibitor can somehow reverse accumulation of acetyl H3, aberrant cell cycle activity, DNA damage, neuronal loss, as well as impaired learning/memory and motor deficits. Overall, the manuscript is well organized with some novel insights of the implication of TDP43/HDAC1 connection in FTLD. The rescue effects of the HDAC1 specific activator also looks rather promising. However, my major concern is that many of the data presented in this study are correlative observations, and many of the results were over-interpreted. Several points need to be clarified to strengthen the mechanisms underlying FTLD-TDP pathologies to support the conclusion of this study.:

Major points:

1. Linear regression analysis data in Fig1C and S1E only reveals that there are more Ki67+ cells when there are more TDP43 mislocalised cells in FTLD-TDP Tg mice. It cannot lead to the assumption that "TDP-43 proteinopathies precede the onset of cell cycle aberrance". Actually, from Fig S1D, some of the Ki67+ cells does not show TDP43 mislocalisation. It requires a time serial examination to show that TDP43 mislocalisation is indeed occurred before the presence of Ki67+ cells

The same for Fig 2G and S2C, the correlations cannot make the conclusion that "cells with aberrant cell cycle activity were highly susceptible to DNA damage". I personally feel it might be better to reorganize Fig1,2, S1,S2,S3, as they look quite redundant.

The sentence in the abstract "we found that aberrant cell cycle activity and DNA damage are key pathogenic factors in FTLD-TDP transgenic" should also be toned down. This is no direct evidence

to reflect this, as the data are simply correlative at this stage.

This could be applied to several of the figure legends of the title of Figures 1, 2, and 3. All these data are correlative, without a design for the cause and consequence experiment. Their data are insufficient to support their conclusion.

To provide a stronger link that HDAC1 mislocalisation is associated with DNA damage and cell cycle activation, I suggest to check the whether cells with HDAC1 mislocalization (Fig 4) also show Ki67+/γH2AX+ signals.

2. From the authors' previous study {Tsai, 2010 #654}, the mice already manifest learning/memory deficits at an earlier age (2 months-old) with changes of learning/memory-associated proteins and obvious gliosis. Does nuclear HDAC1 activity already decreased at 2 months-old? Additionally, given that DNA damage and aberrant cell cycle activation were also observed in non-neurons, did the authors observed amelioration of gliosis after HDAC1 treatment?

3. Fig3b: is there a non-overexposed blot to show a reduction of TDP43 in the nucleus?

4. In figure 8, are those Ki67+/γH2AX+ signals in cells with TDP43 or HDAC1 mislocalisation or HDAC1 in the patients with FTLT-TDP? The color contrast in the higher magnifications does not faithfully recapitulate the original intensity in figure 8a and d. Also please describe the TDP43 mutations carried by individual patients in Method section.

5. Given that the domain of HDAC1 to interact with TDP43 shares high sequence homology with HDAC2, did the authors check the involvement of HDAC2? Please add a discussion about the possibility.

Minor points:

1. Many scale bars are missing in figures.

2. The quantification method in each group needs to be better elaborated.

3. IRB statements about using human patient samples should be enclosed in the method section.

4. In page 9, the sentence "The function of HDAC1 in modulating cell cycle activity and DNA repair in the brain is well characterized, as is its deregulation in various neurodegenerative conditions (Kim et al, 2008; Miller et al, 2010)." needs to be revised with correct grammar.

Referee #3 (Comments on Novelty/Model System for Author):

See main remarks below

Referee #3 (Remarks for Author):

In this manuscript, the authors have observed aberrant cell cycle activity and DNA damage in affected neurons in a transgenic model for frontotemporal lobar degeneration, caused by overexpression of TAR DNA-binding protein 43 (TDP-43). Delving into the mechanisms for this, the authors attribute the cell cycle activity and DNA damage to a direct interaction between TDP-43 and histone deacetylase 1 (HDAC1), which causes HDAC1 to mis-localize to the cytoplasm, in effect inactivating HDAC1. The authors show that pharmacological activation of HDAC1 in the

TDP-43 model can improve the behavioral deficits and ameliorate the pathological features (cell cycle activation and DNA damage).

TDP-43 is an extremely important protein implicated in the pathology of frontotemporal dementia, ALS, and beyond, and understanding the mechanisms of how it causes neurodegeneration has important far reaching implications in the neurodegeneration field. The proposed mechanism, the sequestration of HDAC1 into cytosol via direct interaction with TDP-43 leading to cell cycle reentry and DNA damage, is highly novel. The demonstration of in vivo effects of HDAC1 activation has very intriguing clinical implications as well. Overall, the study is extremely interesting with a potential for broad impact. The paper is well written and easy to follow.

On the other hand, while the overall body of experimental work is impressive, I feel the central mechanism needs more experimental substantiation. The sequestration of HDAC1 into the cytosol by TDP-43 is the central mechanism of the study but I feel more experiments or some additional controls are required to fully demonstrate this. The cytosolic localization of TDP-43 is clearly shown with proper magnification in Figure 1A, however, Figure 4A, which is the most important micrograph in the entire study, is zoomed out too much to tell anything. Is the TDP-43 and HDAC1 co-localized in the cytoplasm at later timepoints, as the authors describe in the text? Zoomed in images where we can tell whether TDP-43 cytoplasmic inclusions also hold HDAC1 - and quantifications of the proportions of neurons where one sees this - this sort of thing is required. Cell fractionation in 3B is a nice addition but the effects are very subtle and hard to make out differences, careful quantification of the bands and averages across multiple animals should be shown. Finally, can the authors find any evidence of cytosolic sequestration of HDAC1 in frontotemporal lobar degeneration patient brain samples, and can this be quantified? This is a key issue that needs to be addressed for the study to be suitable for publication in EMBO Molecular Medicine.

Some more minor comments are as follows:

- 1) in Fig 5B, the HDAC1 fragments themselves should also be shown in the input and IP samples.
- 2) Many details are missing regarding quantification of micrographs such as in 1B, 2A, 2C, etc. While n=5 mice is a good number, how many fields were counted, how many total cells are counted?
- 3) In page 10, P21, E2F1, and acetyl histone H3 are referred to as substrates of HDAC1, a wildly inaccurate statement. While acetyl-histone H3 is indeed a substrate, p21 and E2F1 should be referred to as targets of HDAC1 as they are not direct substrates but instead transcriptionally affected via histone deacetylation.
- 4) At least for a few readouts such as H2AX, non-affected brain regions of the mouse model should be shown and quantified as a negative control.

The authors would like to thank the great effort of the reviewers and editor for their helpful comments that help to improve the manuscript. Our replies to the reviewers' comments and revisions of the manuscript are described in detail below (in the order of comments by reviewers #1, #2 and #3, respectively). The changed sections in the final manuscript have been highlighted for the reviewers by underlining them by using the track changes mode in MS Word.

Reviewer #1

Major points:

1. The authors state that certain observed alterations (i.e., DNA damage, cell cycle markers, etc.) must play a role in the pathogenesis of the disease. These statements must be thoroughly corrected. The only alteration that they can claim to play a disease role is the decline in HDAC1 activity, if the effects of compound 5104434 are specifically HDAC1 activation.

Response

We thank the reviewer for the professional and critical questions. To answer your question about certain observations that we claim are involved in the pathogenesis of the disease, we have toned down our statement to a more euphemistic one in our manuscript, showing that with the data of the specific HDAC1 inhibitor 5104434, only the decline of HDAC1 activity is directly participated in the disease pathogenesis.

(Page 2, Abstract) (Page 9, section 1,2) (Page 16, section 1)

2. It appears that the only reference to compound 5104434 is an international patent held by the corresponding author since 2010 (WO2010011318). The effects of 5104434 on the mouse are the clincher of this work, because they link together the other observations. More information and data on this compound (potency, specificity, bioavailability, safety, etc.) must be provided, and potential conflicts of interests must be clearly stated.

Response

(A) The major point the reviewer mentioned was to provide more information about the compound 5104434. We must explain clearly here that the compound's international patent is not held by us, but belongs to Dr. Li-Huei Tsai, the director

of Picower Institute for Learning and Memory, MIT.

- (B) For this compound, **I, Kuen-Jer Tsai, hereby disclose that our team has no conflicts of interest, nor do we have any patent requirements**, which we have addressed in our manuscript as well. (Page 30, section 3)
- (C) To answer the issue about the compound's safety, we present the following data, including cell viability under compound 5104434 treatment in the 293T cell line (in the manuscript Fig. EV5A) and detected levels of aspartate transaminase (AST), alanine aminotransferase (ALT), creatinine, and lactic dehydrogenase (LDH) in the serum of both wildtype and Tg mice with 5104434 treatment (n=10), which has been updated in the manuscript Fig. EV5B. We asked the original company (ChemBridge Corporation; San Diego, CA, USA) for certain information; they only provided us this link: <https://www.hit2lead.com/screening-compounds/5104434> and a safety data sheet which showed little information (attached with the letter).
3. A genetic approach to confirm the requirement of HDAC1 for the described effects of compound 5104434 would be reassuring.

Response

According to the reviewer's suggestion to provide a genetic approach to confirm the requirement of HDAC1 for compound 5104434, a construct expressing an enzymatically inactive HDAC1 mutant was used (H140/141A)(Zupkovitz et al, 2006). We show that HDAC1 activity is increased in the groups of endogenous (Vec) and over-expressed wild-type full length HDAC1 (HDAC1 FL) under 5104434 treatment, but not in the group of mutant HDAC1 (as shown in Figure below). This result indicating that the histone deacetylase activity of HDAC1 is required for the effects of 5104434 we showed in the manuscript.

Relative HDAC1 activity in the SH-SY5Y cell line exotic expressed wildtype or mutant HDAC1 under compound 5104434 treatment. Data were analyzed by multiple comparisons and are represented as mean \pm SEM (N=3). # p < 0.05, ##### or **** p < 0.0001 by multiple comparison.

4. Compound 5104434 does not modify the localization of HDAC1 or TDP-43 mislocalization, but presumably increases the activity of residual nuclear HDAC1. However, it is necessary to define whether it affects neuronal pathology, not just in terms of markers of DNA damage and cell cycle, but also of neuroinflammation, gliosis, etc.

Response

In response to the reviewer's recommendation, as compound 5104434 increases the activity of residual nuclear HDAC1, we performed immunofluorescence staining of GFAP to show gliosis in the brain. The new graph has been updated in the manuscript Fig. 7G. The number of GFAP⁺ cells increased in the hippocampal region of Tg + vehicle group of mice, compared to WT + vehicle group of mice, where the number of GFAP⁺ cells decreased under compound 5104434 treatment. Treatment with compound 5104434 indeed reduced gliosis. This result indicates that compound 5104434 can ameliorate neuroinflammation and gliosis induced by TDP-43 pathology.

5. The co-IP experiments in figure 5 must be quantified, especially 5C.

Response

In response to the reviewer's advice, we have increased the sample sizes (N=5 mice per group) and quantified the co-IP data in Fig. 5. Please see the latest version of

our manuscript.

Reviewer #2

Major points:

1. Human sample details need to be described more in details

Response

We thank the reviewer for their professional and critical questions. We have re-addressed the human sample details; please see the latest version of our manuscript. (Page 28, section 2)

2. My major concern is that many of the data presented in this study are correlative observations, and many of the results were over-interpreted.

Response

We have removed the misleading assumption and conclusion and change our statement to a more euphemistic one in the manuscript according to the reviewers' advice.

(Page 2, Abstract)(Page 9, section 1, 2) (Page 16, section 1)

3. Linear regression analysis data in Fig1C and S1E only reveals that there are more Ki67+ cells when there are more TDP43 mislocalized cells in FTLD-TDP Tg mice. It cannot lead to the assumption that "TDP-43 proteinopathies precede the onset of cell cycle aberrance". Actually, from Fig S1D, some of the Ki67+ cells does not show TDP43 mislocalization. It requires a time serial examination to show that TDP43 mislocalization is indeed occurred before the presence of Ki67+ cells

Response

To answer the major question about providing a serial time examination, we have co-stained TDP-43 and Ki67 in the brain of 2-month-old Tg mice. At that time there is no TDP-43 mislocalization, and Ki67 is not expressed in the frontal cortices (Fig. EV1). According to our previous study, there is no TDP-43 mislocalization at the age of 2 months either (Tsai 2010). Therefore, although we cannot provide data showing a transient stage of mislocalized TDP-43 and no Ki67 data exist to support the previous assumption, we can assume the observation of cell cycle aberrance is due to TDP-43 proteinopathies.

4. The same for Fig 2G and S2C, the correlations cannot make the conclusion that "cells with aberrant cell cycle activity were highly susceptible to DNA damage". I personally feel it might be better to reorganize Fig1,2, S1,S2,S3, as they look quite redundant.

Response

- (A) We have corrected the sentence to “cells/ neurons with aberrant cell cycle activity **also suffered** DNA damage”. (Page 9, section 2)
- (B) We have reorganized Fig. 1, 2, and Fig.S1-S3 as well. Please see the latest version of our manuscript. (Page 7-9, section 1)

5. The sentence in the abstract "we found that aberrant cell cycle activity and DNA damage are key pathogenic factors in FTLD-TDP transgenic" should also be toned down. This is no direct evidence to reflect this, as the data are simply correlative at this stage. This could be applied to several of the figure legends of the title of Figures 1, 2, and 3. All these data are correlative, without a design for the cause and consequence experiment. Their data are insufficient to support their conclusion.

Response

We have corrected the sentence to “aberrant cell cycle activity and DNA damage are **important** pathogenic factors”. (Page 2, Abstract) We have also changed the figure legend title in Fig. 1-3 as the reviewer’s suggestion.

6. To provide a stronger link that HDAC1 mislocalization is associated with DNA damage and cell cycle activation, I suggest to check whether the cells with HDAC1 mislocalization (Fig 4) also show Ki67⁺/ γ H2AX⁺ signals.

Response

This data has been added in the manuscript in Fig. 4B. We have showed the IF staining and quantification of cell numbers with γ H2AX⁺ and HDAC1 mislocalization in the brain of both wildtype and FTLD-TDP Tg mice. Please see the latest version of our manuscript.

7. Does nuclear HDAC1 activity already decreased at 2 months-old? Additionally,

given that DNA damage and aberrant cell cycle activation were also observed in non-neurons, did the authors observed amelioration of gliosis after HDAC1 treatment?

Response

(A) To answer the question about whether HDAC1 activity already decreases in 2-month-old Tg mouse brains, we checked the histone deacetylase activity of HDAC1. There is no difference in the HDAC1 activity between WT and 2 month-old Tg mice (as shown in Figure below). Meanwhile, TDP-43 is not mislocalized at this stage (Tsai et al, 2010), so we assume that there is no HDAC1 mislocalization.

Relative HDAC1 activity in the forebrain of 2-month-old WT or Tg mice. Data were analyzed by *t*-test and are represented as mean \pm SEM (N=3 mice per group).

(B) To address the question whether gliosis has been ameliorated after compound 5104434 treatment, we performed immunofluorescence staining of GFAP to show gliosis in the brain. The new graph has been updated in the manuscript Fig. 7G. The number of GFAP⁺ cells is increased in the hippocampal region of Tg + vehicle group of mice compared to the WT + vehicle group of mice, where the number of GFAP⁺ cells is decreased under compound 5104434 treatment. Treatment of compound 5104434 indeed reduced gliosis. This result indicates that compound 5104434 can ameliorate neuroinflammation and gliosis induced by TDP-43 pathology.

8. Fig3b: is there a non-overexposed blot to show a reduction of TDP43 in the nucleus?

Response

We have changed and improved the data quality of Fig. 3B. Please see the latest

version of our manuscript.

9. In figure 8, are those Ki67⁺/ γ H2AX⁺ signals in cells with TDP43 or HDAC1 mislocalization or HDAC1 in the patients with FTLD-TDP? The color contrast in the higher magnifications does not faithfully recapitulate the original intensity in figure 8a and d. Also please describe the TDP43 mutations carried by individual patients in Method section.

Response

- (A) In response to the reviewer's questions, according to Fig. 8G, although we don't have co-staining of TDP-43 with Ki67⁺ or H2AX⁺ data in the patient samples, some of the Ki67⁺ and H2AX⁺ cells are most likely to be TDP-43 and HDAC1 mislocalized cells as well.
- (B) We have also rearranged the color contrast in Fig. 8A and 8D.
- (C) The brain samples from FTLD-TDP patients were kindly offered by Dr. Jin, but there is no genetic confirmation of the TDP-43 mutations they carried.

10. Given that the domain of HDAC1 to interact with TDP43 shares high sequence homology with HDAC2, did the authors check the involvement of HDAC2 ? Please add a discussion about the possibility.

Response

According to the suggestion of checking the involvement of HDAC2, we have done the following experiment and prepared the following paragraph in the Discussion: "We have also performed the nuclear HDAC2 activity assay. Despite that HDAC1 and HDAC2 share highly conserved N-terminal regions including the protein-protein interaction domain (1-52) and the catalytic domain (12-322), there is no difference in the HDAC2 activity between WT and Tg mice (as shown in Figure below). Therefore, it is more likely that HDAC1, but not HDAC2, is affected by nuclear TDP-43 mislocalization, and thus contributes to TDP-43 proteinopathies." (Page 20, section 2)

Relative HDAC2 activity in the mouse brain of 6-month-old WT or FTLD-TDP Tg mice. Data were analyzed by *t*-test and are represented as mean \pm SEM (N=5 mice per group).

11. Minor points:

1. Many scale bars are missing in figures.
2. The quantification method in each group needs to be better elaborated.
3. IRB statements about using human patient samples should be enclosed in the method section.
4. In page 9, the sentence "The function of HDAC1 in modulating cell cycle activity and DNA repair in the brain is well characterized, as is its deregulation in various neurodegenerative conditions (Kim et al, 2008; Miller et al, 2010). " needs to be revised with correct grammar.

Response

- (A) For the minor points that the reviewer reminded us about, we have checked the scale bars in all figures. (Fig. 1, 2, 4, 8)
- (B) We were more specific regarding the quantification methods and the IRB statements. (Page 23, section 1) (Page 28, section 2)
- (C) We have also rewritten the sentence in page 9 to " The function of HDAC1 in modulating cell cycle activity and DNA repair in the brain is well-characterized, as **the dysregulation of HDAC1 is seen** in various neurodegenerative conditions". We have examined the grammar more carefully in our manuscript. Please see the latest version of our manuscript. (Page 10, section 1)

Reviewer #3

Major points:

1. The cytosolic localization of TDP-43 is clearly shown with proper magnification in Figure 1A, however, Figure 4A, which is the most important micrograph in the entire study, is zoomed out too much to tell anything. Is the TDP-43 and HDAC1 co-localized in the cytoplasm at later timepoints, as the authors describe in the text? Zoomed in images where we can tell whether TDP-43 cytoplasmic inclusions also hold HDAC1 - and quantifications of the proportions of neurons where one sees this - this sort of thing is required.

Response

We thank the reviewer for their professional and critical questions. We have zoomed in on the images in Fig. 4A and one can clearly see TDP-43 and HDAC1 co-localized in the cytoplasm in 6-month-old and 12-month-old Tg mouse brains, but not in the early stage of Tg mice or late stage of WT mice.

2. Cell fractionation in 3B is a nice addition but the effects are very subtle and hard to make out differences, careful quantification of the bands and averages across multiple animals should be shown.

Response

We have changed and improved the data quality of Fig. 3B, and carefully quantified the levels of TDP-43 in both cytosolic and nuclear samples of multiple animals (n=5 per group).

3. Can the authors find any evidence of cytosolic sequestration of HDAC1 in frontotemporal lobar degeneration patient brain samples, and can this be quantified?

Response

In Fig. 8A, we showed cytosolic sequestration of HDAC1 in the brain of FTLTDP patients, the levels of mislocalized HDAC1 were also measured in the **RIPA-insoluble/urea-soluble protein fractions** of patients' brain samples (Fig. 8G), and we have also quantified the immunoblotting (Fig. 8H).

4. in Fig 5B, the HDAC1 fragments themselves should also be shown in the input and IP samples.

Response

Regarding some minor comments by the reviewer, we have added the input figure in Fig. 5B.

5. Many details are missing regarding quantification of micrographs such as in 1B,2A, 2C, etc. While n=5 mice is a good number, how many fields were counted, how many total cells are counted?

Response

We were more specific regarding the quantification methods of the micrographs. The IF staining from Fig. 1 and 2 are collected from n= 9 sections per mouse, N = 5 mice per group, and the data are presented as mean \pm SEM. (Page 23, section 1 and in other figure legend)

6. In page 10, P21, E2F1, and acetyl histone H3 are referred to as substrates of HDAC1, a wildly inaccurate statement. While acetyl-histone H3 is indeed a substrate, p21 and E2F1 should be referred to as targets of HDAC1 as they are not direct substrates but instead transcriptionally affected via histone deacetylation.

Response

We have written more carefully with the words “substrate” and “targets”, *et cetera*. Please see the latest version of our manuscript. (Page 10, section 1) (Page 14, section 2)

7. At least for a few readouts such as H2AX, non-affected brain regions of the mouse model should be shown and quantified as a negative control.

Response

According to the reviewer’s advice, we have added γ H2AX staining of the cerebella of 1-year-old Tg mice, which shows no signal at all (Fig. S1). Without

further quantification, this result should be seen as an negative control.

REFERENCE

Zupkovitz G, Tischler J, Posch M, Sadzak I, Ramsauer K, Egger G, Grausenburger R, Schweifer N, Chiocca S, Decker T et al (2006) Negative and positive regulation of gene expression by mouse histone deacetylase 1. *Mol Cell Biol* 26: 7913-7928

10th Sep 2019

Dear Prof. Tsai,

Thank you for the submission of your revised manuscript to EMBO Molecular Medicine. We have now received the enclosed report from the referees who were asked to re-assess it. As you will see the referees are now overall supportive and I am pleased to inform you that we will be able to accept your manuscript pending the following amendments:

1. Please address the comments of referee #2 (figure annotation, wording correction, discussion and scale bar)
2. Figures: Please add scale bars to Fig2A and Fig4A.
3. In the main manuscript file, please do the following:
 - Accept all changes and remove the red color font
 - check the figure callouts in the main article. Currently, Fig 4C is called out in the main article, but not provided in the figure panel.
 - the abbreviation section should be deleted and individual abbreviations should be incorporated in the main text as they are first seen.
 - indicate in legends exact n= and exact p= values, not a range, along with the statistical test used. Some people found that to keep the figures clear, providing an Appendix table S with all exact p-values was preferable. You are welcome to do this if you want to.
 - in Materials and Methods, provide the antibody dilutions that were used for each antibody
 - in Materials and Methods (as well as checklist), for animal work, confirm that all experiments were performed in accordance with relevant guidelines and regulations. The manuscript must include a statement in the Materials and Methods identifying the institutional and/or licensing committee approving the experiments. Gender, age and genetic background must be indicated, along with housing conditions.
 - in Materials and Methods, include a statement that informed consent was obtained from all subjects and that the experiments conformed to the principles set out in the WMA Declaration of Helsinki and the Department of Health and Human Services Belmont Report.
 - remove "data not shown". As per our guidelines, on "Unpublished Data" the journal does not permit citation of "data not shown". All data referred to in the paper should be displayed in the main or Expanded View figures. "Unpublished observations" may be referred to in exceptional cases, where these are data peripheral to the major message of the paper and are intended to form part of a future or separate study, the names of the persons that reported the observation should be listed in brackets. Personal communications (Author name(s), personal communications) must be authorized in writing by those involved, and the authorization sent to the editorial office at time of submission.
4. Figures: many of the IF staining panels appear empty (such as Fig 1A, Fig 2A, B etc). Please adjust contrast so that the background can be seen (even if is very faint).
5. The Data Availability section is meant for sharing large scale datasets. Since this is not the case here, please remove the current "Data Availability" section.
6. In the online submission system, please enter the grant number information (including all grants)

that is currently listed in the acknowledgements section in the main article.

7. The Paper Explained: EMBO Molecular Medicine articles are accompanied by a summary of the articles to emphasize the major findings in the paper and their medical implications for the non-specialist reader. Please provide a draft summary of your article highlighting

- a. the medical issue you are addressing (heading: PROBLEM)
- b. the results obtained (heading: RESULTS)
- c. their clinical impact (heading: IMPACT).
- d. This may be edited to ensure that readers understand the significance and context of the research. Please refer to any of our published articles for an example.

8. We would also encourage you to include the source data for figure panels that show essential data. Numerical data should be provided as individual .xls or .csv files (including a tab describing the data). For blots or microscopy, uncropped images should be submitted (using a zip archive if multiple images need to be supplied for one panel). Additional information on source data and instruction on how to label the files are available at

<https://www.embopress.org/page/journal/17574684/authorguide#sourcedata>

9. As part of the EMBO Publications transparent editorial process initiative (see our Editorial at <http://embomolmed.embopress.org/content/2/9/329>), EMBO Molecular Medicine will publish online a Review Process File (RPF) to accompany accepted manuscripts.

In the event of acceptance, this file will be published in conjunction with your paper and will include the anonymous referee reports, your point-by-point response and all pertinent correspondence relating to the manuscript. Please let me know if you agree with this.

10. For More Information: There is space at the end of each article to list relevant web links for further consultation by our readers. Could you identify some relevant ones and provide such information as well? Some examples are patient associations, relevant databases, OMIM/proteins/genes links, author's websites, etc...

11. I have slightly modified the Synopsis text. Could you please let me know if it is fine like this or if you would like to introduce further changes?

****Synopsis**:**

TDP-43 proteinopathies cause pathogenesis through inducing cytosolic mislocalization of HDAC1. Pharmacological recovery of HDAC1 activity in FTLD-TDP Tg mice can improve cognitive and motor function, also attenuate aberrant cell cycle activity, DNA damage and neuronal death.

- Aberrant cell cycle activity and DNA damage are found in frontal cortices of both FTLD-TDP transgenic (Tg) mice and FTLD-patients.
- TDP-43 interacts with HDAC1 and traps it in cytosolic inclusions during the pathogenesis of TDP-43 proteinopathies.
- TDP-43 proteinopathies may play an essential role in reducing nuclear levels and activity of HDAC1.
- Increased HDAC1 activity ameliorates the cognitive and motor function of Tg mice, also reduces DNA damage and neuronal loss.

Please submit your revised manuscript within two weeks. I look forward to seeing a revised version of your manuscript as soon as possible.

Yours sincerely,
Jingyi Hou

Jingyi Hou
Editor
EMBO Molecular Medicine

*** Instructions to submit your revised manuscript ***

To submit your manuscript, please follow this link:

Link Not Available

- 1) a .doc formatted version of the manuscript text (including Figure legends and tables)
- 2) Separate figure files*
- 3) supplemental information as Expanded View and/or Appendix. Please carefully check the authors guidelines for formatting Expanded view and Appendix figures and tables at <https://www.embopress.org/page/journal/17574684/authorguide#expandedview>
- 4) a letter INCLUDING the reviewer's reports and your detailed responses to their comments (as Word file).

5) The paper explained: EMBO Molecular Medicine articles are accompanied by a summary of the articles to emphasize the major findings in the paper and their medical implications for the non-specialist reader. Please provide a draft summary of your article highlighting

6) For more information: There is space at the end of each article to list relevant web links for further consultation by our readers. Could you identify some relevant ones and provide such information as well? Some examples are patient associations, relevant databases, OMIM/proteins/genes links, author's websites, etc...

7) Author contributions: the contribution of every author must be detailed in a separate section.

8) EMBO Molecular Medicine now requires a complete author checklist (<https://www.embopress.org/page/journal/17574684/authorguide>) to be submitted with all revised manuscripts. Please use the checklist as guideline for the sort of information we need WITHIN the manuscript. The checklist should only be filled with page numbers where the information can be found. This is particularly important for animal reporting, antibody dilutions (missing) and exact values and n that should be indicated instead of a range.

9) Every published paper now includes a 'Synopsis' to further enhance discoverability. Synopses are displayed on the journal webpage and are freely accessible to all readers. They include a short stand first (maximum of 300 characters, including space) as well as 2-5 one sentence bullet points that summarise the paper. Please write the bullet points to summarise the key NEW findings. They should be designed to be complementary to the abstract - i.e. not repeat the same text. We encourage inclusion of key acronyms and quantitative information (maximum of 30 words / bullet point). Please use the passive voice. Please attach these in a separate file or send them by email, we will incorporate them accordingly.

You are also welcome to suggest a striking image or visual abstract to illustrate your article. If you do please provide a jpeg file 550 px-wide x 400-px high.

10) A Conflict of Interest statement should be provided in the main text

11) Please note that we now mandate that all corresponding authors list an ORCID digital identifier. This takes <90 seconds to complete. We encourage all authors to supply an ORCID identifier, which will be linked to their name for unambiguous name identification.

Currently, our records indicate that the ORCID for your account is 0000-0002-2170-9735.

Link Not Available

12) The system will prompt you to fill in your funding and payment information. This will allow Wiley to send you a quote for the article processing charge (APC) in case of acceptance. This quote takes into account any reduction or fee waivers that you may be eligible for. Authors do not need to pay any fees before their manuscript is accepted and transferred to our publisher.

Photos 400-800 DPI

*Additional important information regarding figures and illustrations can be found at <http://bit.ly/EMBOPressFigurePreparationGuideline>

The system will prompt you to fill in your funding and payment information. This will allow Wiley to send you a quote for the article processing charge (APC) in case of acceptance. This quote takes into account any reduction or fee waivers that you may be eligible for. Authors do not need to pay any fees before their manuscript is accepted and transferred to our publisher.

***** Reviewer's comments *****

Referee #1 (Remarks for Author):

The authors have addressed most of the concerns raised. The majority of the findings are convincing and support the interpretation that HDAC1 is associated with the pathogenic process, although causality remains uncertain at this stage.

Referee #2 (Remarks for Author):

The authors carried out additional experiments and reorganized the main text to address our concerns. The current version has greatly improved to support HDAC1 as a potential therapeutic target for TDP43 proteinopathy through reducing aberrant cell cycle and DNA damage in frontal cortices.

A few minor points are suggested below for the authors to address in the text.

1. As the FTLD-TDP Tg mice mice progressively exhibit cognitive deficits starting from 2 months of age 1, a stage when no TDP43/HDAC1 mislocalization appears. Does it mean that HDAC1 dysregulation represents a second pathogenic wave contributing to cognitive deficits? We suggest the authors to include this issue in the discussion.

2. Page 8: "we investigated the relationship between TDP-43 proteinopathies and DNA damage with IF staining TDP-43 of and γ H2AX in 6-mon-old FTLD-TDP Tg and WT mice." Please correct the underlined wordings.

3. Page 11: "At the age of 12 months, the co-mislocalization of TDP-43 and HDAC1 had progressed in the cells of FTLD-TDP Tg mice but not in the cells of age-matched WT mice, which revealed an age-dependent effect (Fig. 4B). We also confirm that more γ H2AX expressed in the nucleus when cells undergo HDAC1 mislocalization in the frontal cortex of 12-mon-old FTLD-TDP Tg mice (Fig. 4C, left graph), but not in the cells of age-matched WT mice (Fig. 4C, right histogram)." Please correct the underlined figure annotations.

4. The scale bar value is missing in Figure legend of 7E~7G.

Reference:

1 Tsai, K. J. et al. Elevated expression of TDP-43 in the forebrain of mice is sufficient to cause neurological and pathological phenotypes mimicking FTLD-U. *The Journal of experimental medicine* 207, 1661-1673, doi:10.1084/jem.20092164 (2010).

Referee #3 (Comments on Novelty/Model System for Author):

The authors have addressed all of my concerns in a satisfactory manner. In particular, verification of cytosolic HDAC1 sequestration in FTLD-TDP patient samples, and improved quantitative analyses of HDAC1 and TDP-43 localization throughout the manuscript, improves the overall impact and technical quality of the study. In my opinion the study is suitable for publication for *EMBO Molecular Medicine* in its current state.

Referee #3 (Remarks for Author):

The authors have addressed all of my concerns in a satisfactory manner. In particular, verification of cytosolic HDAC1 sequestration in FTLD-TDP patient samples, and improved quantitative analyses of HDAC1 and TDP-43 localization throughout the manuscript, improves the overall impact and technical quality of the study. In my opinion the study is suitable for publication for *EMBO Molecular Medicine* in its current state.

Amendments

The authors would like to thank the great effort of the editor and reviewers for their professional and helpful comments that help to improve the manuscript. Our replies to the comments and revisions of the manuscript are described in detail below (in the order of comments by editor and reviewers #2, respectively).

1. Please address the comments of referee #2 (figure annotation, wording correction, discussion and scale bar)

Response

We have addressed all the comments of referee #2, including figure annotation, wording correction, discussion and scale bar. We appreciate his comments to improve the quality of our manuscript.

2. Figures: Please add scale bars to Fig2A and Fig4A.

Response

These parts have been added and corrected.

3. In the main manuscript file, please do the following:
 - Accept all changes and remove the red color font

Response

This part has been done.

- Check the figure callouts in the main article. Currently, Fig 4C is called out in the main article, but not provided in the figure panel.

Response

We apologized for the mistake, this has been corrected to Fig. 4B.

- The abbreviation section should be deleted and individual abbreviations should be incorporated in the main text as they are first seen.

Response

This part has been done.

- Indicate in legends exact n= and exact p= values, not a range, along with the statistical test used. Some people found that to keep the figures clear, providing an Appendix table S with all exact p-values was preferable. You are welcome to do this if you want to.

Response

We have corrected and put exact p-values in each of the figure legends.

- In Materials and Methods, provide the antibody dilutions that were used for each antibody.

Response

We have added exact dilutions ratio of all antibodies in Materials and Methods.

- In Materials and Methods (as well as checklist), for animal work, confirm that all experiments were performed in accordance with relevant guidelines and regulations. The manuscript must include a statement in the Materials and Methods identifying the institutional and/or licensing committee approving the experiments. Gender, age and genetic background must be indicated, along with housing conditions.

Response

The statement and other information such as mouse gender have been added into the Materials and Methods.

- In Materials and Methods, include a statement that informed consent was obtained from all subjects and that the experiments conformed to the principles set out in the WMA Declaration of Helsinki and the Department of Health and Human Services Belmont Report.

Response

The statement has been added into the Materials and Methods.

- Remove "data not shown". As per our guidelines, on "Unpublished Data" the journal does not permit citation of "data not shown". All data referred to in the paper should be displayed in the main or Expanded View figures. "Unpublished observations" may be referred to in exceptional cases, where these are data

peripheral to the major message of the paper and are intended to form part of a future or separate study, the names of the persons that reported the observation should be listed in brackets. Personal communications (Author name(s), personal communications) must be authorized in writing by those involved, and the authorization sent to the editorial office at time of submission.

Response

We have change the word "data not shown" into "Unpublished observations", and adding the name of the person who did the experiment within it, who is one of our authors.

4. Figures: many of the IF staining panels appear empty (such as Fig 1A, Fig 2A, B etc). Please adjust contrast so that the background can be seen (even if is very faint).

Response

We have adjusted most of the figure, but some of them are rarely empty. There's no background of any signal.

5. The Data Availability section is meant for sharing large scale datasets. Since this is not the case here, please remove the current "Data Availability" section.

Response

The "Data Availability" section has been removed.

6. In the online submission system, please enter the grant number information (including all grants) that is currently listed in the acknowledgements section in the main article.

Response

This part has been added.

7. The Paper Explained: EMBO Molecular Medicine articles are accompanied by a summary of the articles to emphasize the major findings in the paper and their medical implications for the non-specialist reader. Please provide a draft summary of your article highlighting
 - a. the medical issue you are addressing (heading: PROBLEM)

- b. the results obtained (heading: RESULTS)
- c. their clinical impact (heading: IMPACT).
- d. This may be edited to ensure that readers understand the significance and context of the research. Please refer to any of our published articles for an example.

Response

- a. TAR DNA-binding protein 43 (TDP-43) has been implicated in frontotemporal lobar degeneration (FTLD). Histone deacetylase 1 (HDAC1) is involved in DNA repair and neuroprotection in numerous neurodegenerative diseases. However, the pathological mechanisms of FTLD underlying TDP-43 proteinopathies are unclear. The issue with TDP-43 and HDAC1 is addressed by the paper.
 - b. The results demonstrated cell cycle aberrance and DNA damage are involved in the degenerative progress of FTLD and investigated the role of HDAC1 in TDP-43 proteinopathies. This hypothesis is consistent with nonclinical and clinical findings currently in the paper.
 - c. The paper proposes a framework for the generation of new hypotheses and the conduct of additional studies. Examples include further understanding: HDAC1 deregulation is involved in the pathogenesis of TDP-43 proteinopathies, and HDAC1 is a potential target for therapeutic interventions in FTLD. By restoring HDAC1 activity may be a feasible approach to treating FTLD.
8. We would also encourage you to include the source data for figure panels that show essential data. Numerical data should be provided as individual .xls or .csv files (including a tab describing the data). For blots or microscopy, uncropped images should be submitted (using a zip archive if multiple images need to be supplied for one panel). Additional information on source data and instruction on how to label the files are available at
<https://www.embopress.org/page/journal/17574684/authorguide#sourcedata>

Response

We have organized the files contained the source data and submitted to the system.

- 9. As part of the EMBO Publications transparent editorial process initiative (see our Editorial at <http://embomolmed.embopress.org/content/2/9/329>), EMBO Molecular Medicine will publish online a Review Process File (RPF) to accompany accepted manuscripts.

In the event of acceptance, this file will be published in conjunction with your paper and will include the anonymous referee reports, your point-by-point response and all pertinent correspondence relating to the manuscript. Please let me know if you agree with this.

Response

We are very glad to accept this policy.

10. For More Information: There is space at the end of each article to list relevant web links for further consultation by our readers. Could you identify some relevant ones and provide such information as well? Some examples are patient associations, relevant databases, OMIM/proteins/genes links, author's websites, etc...

Response

We don't have this kind of databases or websites can provide.

11. I have slightly modified the Synopsis text. Could you please let me know if it is fine like this or if you would like to introduce further changes?

We are very glad to accept these modifications.

****Synopsis**:**

TDP-43 proteinopathies cause pathogenesis through inducing cytosolic mislocalization of HDAC1. Pharmacological recovery of HDAC1 activity in FTLN-TDP Tg mice can improve cognitive and motor function, also attenuate aberrant cell cycle activity, DNA damage and neuronal death.

- Aberrant cell cycle activity and DNA damage are found in frontal cortices of both FTLN-TDP transgenic (Tg) mice and FTLN-patients.
- TDP-43 interacts with HDAC1 and traps it in cytosolic inclusions during the pathogenesis of TDP-43 proteinopathies.
- TDP-43 proteinopathies may play an essential role in reducing nuclear levels and activity of HDAC1.
- Increased HDAC1 activity ameliorates the cognitive and motor function of Tg mice, also reduces DNA damage and neuronal loss.

Response

The authors would like to thank the great effort of the editor for her helpful comments that help to improve the quality of manuscript. The synopsis looks fine. We really appreciate her efforts on editing.

Referee #2 (Remarks for Author):

Major points:

1. As the FTLN-TDP Tg mice progressively exhibit cognitive deficits starting from 2 months of age (1), a stage when no TDP43/HDAC1 mislocalization appears. Does it mean that HDAC1 dysregulation represents a second pathogenic wave contributing to cognitive deficits? We suggest the authors to include this issue in the discussion.

Response

Thanks for the reviewer's professional comment. As what the reviewer's observation, HDAC1 dysregulation represents a second pathogenic wave contributing to cognitive deficits. We have included the issue in Discussion (Page 18, lane 7).

2. Page 8: "we investigated the relationship between TDP-43 proteinopathies and DNA damage with IF staining TDP-43 of and γ H2AX in 6-mon-old FTLN-TDP Tg and WT mice." Please correct the underlined wordings.

Response

We have corrected the underlined wordings now in Page 7, lane 11.

3. Page 11: "At the age of 12 months, the co-mislocalization of TDP-43 and HDAC1 had progressed in the cells of FTLN-TDP Tg mice but not in the cells of age-matched WT mice, which revealed an age-dependent effect (Fig. 4B). We also confirm that more γ H2AX expressed in the nucleus when cells undergo HDAC1 mislocalization in the frontal cortex of 12-mon-old FTLN-TDP Tg mice (Fig. 4C, left graph), but not in the cells of age-matched WT mice (Fig. 4C, right histogram)." Please correct the underlined figure annotations.

Response

We have corrected the underlined figure annotations now in Page 9.

4. The scale bar value is missing in Figure legend of 7E~7G.

Response

We have added the scale bar value in Figure legend of 7E~7G.

24th Apr 2020

Dear Prof. Tsai,

We are pleased to inform you that your manuscript is accepted for publication and is now being sent to our publisher to be included in the next available issue of EMBO Molecular Medicine.

We would like to remind you that as part of the EMBO Publications transparent editorial process initiative, EMBO Molecular Medicine will publish a Review Process File online to accompany accepted manuscripts. If you do NOT want the file to be published or would like to exclude figures, please immediately inform the editorial office via e-mail.

Please read below for additional IMPORTANT information regarding your article, its publication and the production process.

Congratulations on your interesting work,

Jingyi Hou

Jingyi Hou
Editor
EMBO Molecular Medicine

Follow us on Twitter @EmboMolMed
Sign up for eTOCs at embopress.org/alertsfeeds

***** Reviewer's comments *****

*** ** IMPORTANT INFORMATION *** **

SPEED OF PUBLICATION

The journal aims for rapid publication of papers, using the advance online publication "Early View" to expedite the process: A properly copy-edited and formatted version will be published as "Early View" after the proofs have been corrected. Please help the Editors and publisher avoid delays by providing e-mail address(es), telephone and fax numbers at which author(s) can be contacted.

Should you be planning a Press Release on your article, please get in contact with embomolmed@wiley.com as early as possible, in order to coordinate publication and release dates.

LICENSE AND PAYMENT:

All articles published in EMBO Molecular Medicine are fully open access: immediately and freely

available to read, download and share.

EMBO Molecular Medicine charges an article processing charge (APC) to cover the publication costs. You, as the corresponding author for this manuscript, should have already received a quote with the article processing fee separately. Please let us know in case this quote has not been received.

Once your article is at Wiley for editorial production you will receive an email from Wiley's Author Services system, which will ask you to log in and will present you with the publication license form for completion. Within the same system the publication fee can be paid by credit card, an invoice, pro forma invoice or purchase order can be requested.

Payment of the publication charge and the signed Open Access Agreement form must be received before the article can be published online.

PROOFS

You will receive the proofs by e-mail approximately 2 weeks after all relevant files have been sent to our Production Office. Please return them within 48 hours and if there should be any problems, please contact the production office at embopressproduction@wiley.com.

Please inform us if there is likely to be any difficulty in reaching you at the above address at that time. Failure to meet our deadlines may result in a delay of publication.

All further communications concerning your paper proofs should quote reference number

EMM-2019-10622-V3

and be directed to the production office at embopressproduction@wiley.com.

Thank you,

Jingyi Hou
Editor
EMBO Molecular Medicine

Corresponding Author Name: Kuen-Jer Tsai
Journal Submitted to: EMBO Molecular Medicine
Manuscript Number: EMM-2019-10622